# Proposed SmartBarrel System for Monitoring and Assessment of Wine Fermentation Processes Using IoT Nose and Tongue Devices

**DOI:** 10.3390/s25133877

**Published:** 2025-06-21

**Authors:** Sotirios Kontogiannis, Meropi Tsoumani, George Kokkonis, Christos Pikridas, Yorgos Kotseridis

**Affiliations:** 1Laboratory Team of Distributed Microcomputer Systems, Department of Mathematics, University of Ioannina, 45110 Ioannina, Greece; ma13045@uoi.gr; 2Department of Information and Electronic Engineering, International Hellenic University, 57001 Thessaloniki, Greece; gkokkonis@ihu.gr; 3School of Rural and Surveying Engineering, Aristotle University of Thessaloniki, 54124 Thessaloniki, Greece; cpik@topo.auth.gr; 4Enology Laboratory, Agricultural University of Athens, Iera Odos 86, Votanikos, 11855 Athens, Greece; ykotseridis@aua.gr

**Keywords:** precision winemaking, decision support systems, IoT, embedded systems, agriculture 4.0, wine fermentation process, fermentation forecasting, deep learning

## Abstract

This paper introduces SmartBarrel, an innovative IoT-based sensory system that monitors and forecasts wine fermentation processes. At the core of SmartBarrel are two compact, attachable devices—the probing nose (E-nose) and the probing tongue (E-tongue), which mount directly onto stainless steel wine tanks. These devices periodically measure key fermentation parameters: the nose monitors gas emissions, while the tongue captures acidity, residual sugar, and color changes. Both utilize low-cost, low-power sensors validated through small-scale fermentation experiments. Beyond the sensory hardware, SmartBarrel includes a robust cloud infrastructure built on open-source Industry 4.0 tools. The system leverages the ThingsBoard platform, supported by a NoSQL Cassandra database, to provide real-time data storage, visualization, and mobile application access. The system also supports adaptive breakpoint alerts and real-time adjustment to the nonlinear dynamics of wine fermentation. The authors developed a novel deep learning model called V-LSTM (Variable-length Long Short-Term Memory) to introduce intelligence to enable predictive analytics. This auto-calibrating architecture supports variable layer depths and cell configurations, enabling accurate forecasting of fermentation metrics. Moreover, the system includes two fuzzy logic modules: a device-level fuzzy controller to estimate alcohol content based on sensor data and a fuzzy encoder that synthetically generates fermentation profiles using a limited set of experimental curves. SmartBarrel experimental results validate the SmartBarrel’s ability to monitor fermentation parameters. Additionally, the implemented models show that the V-LSTM model outperforms existing neural network classifiers and regression models, reducing RMSE loss by at least 45%. Furthermore, the fuzzy alcohol predictor achieved a coefficient of determination ($R^{2}$) of 0.87, enabling reliable alcohol content estimation without direct alcohol sensing.

## 1. Introduction

The continuous growth of the wine market necessitates the development of higher-quality wine products through standardized industrial processes and the use of IoT. Continuous supervision, control, and implementation of standardized intervention procedures during the alcoholic fermentation of must for wine production are central to the Winemaking Industry [1]. With the advancement of the Internet of Things (IoT) and aligned with Industry 4.0 objectives of real-time measuring, cloud storage, cloud computing, and the use of AI to optimize winemaking processes, innovative solutions for IoT Systems, data acquisition and control are being designed and implemented [2,3]. In this direction, this paper proposes using an AI-capable, low-cost wine fermentation monitoring system to develop a prototype wine fermentation tank that continuously records critical parameters. It will integrate low-cost, energy-efficient embedded sensors for real-time data collection, including key oenological measures (e.g., fermentation curve measurements, brix levels), turbidity/clarity via color measurements, and monitoring released gases during fermentation.

In wineries, the industrial process of monitoring alcoholic fermentation typically involves manual sampling and laboratory analysis for quality control and decision-making. Generally, this can be performed no more than twice per day. For large-scale wineries, such an approach is practically infeasible for all fermentation tanks, risking stalled fermentations or quality degradation [4]. Continuous real-time data logging via low-cost in-tank fermentation sensors could significantly improve processes by taking actions on event occurrence, enabling continuous fermentation rate control, enhancing automation levels, and optimizing workforce allocation to higher-priority production tasks [5]. Continuous monitoring via in-tank sensors could streamline fermentation rate control, boost winery automation, and reduce labor overhead, freeing personnel for other production processes. Although embedded sensors (even rudimentary ones) are not novel, their systematic deployment remains restricted, particularly in Greece.

To this extent, the wine industry poses requirements for using dense wine parameter monitoring and device solutions for small-scale wine-fermentation tasks that involve unsupervised control [6]. There is an urgent need for integrated sensor systems capable of routine measurements and micro-scale fermentations in wineries. The use of basic embedded tank sensors has seen limited adoption, primarily due to cost constraints and the complexity of local monitoring technological integration [7]. There is growing interest from wineries, sensor manufacturers, and startups in incorporating innovative, scale-up, automated enological solutions. Existing industrial fermentation control systems provided by Programmable Logic Controllers (PLCs) of time-interval periodical probing and on-demand supervisory control, as well as via local data acquisition (SCADA) interfaces, have achieved initial production automation [8]. However, Industry 4.0 introduced scalability, portability, and autonomous cloud operation principles of telemetry devices, Ubiquitous IoT connectivity, distributed storage, and intelligence [9], which are still ahead. In order to show progress in this direction, dense, real-time multi-attribute monitoring is needed when using cloud-based services and data collection on preferably schemaless NoSQL databases. This way, the end-user integration provided by notification channels and intelligent, automated suggestions using artificial intelligence will offer Industry 4.0 sensory automated processes [9].

Wine fermentation processes supported by systems of low-cost deployment using IoT, extensibility, and cross-platform visualization (via tablets/smartphones) are pivotal. Periodic statistical analysis and AI-driven decision support—including oenological intervention recommendations and final product quality predictions—will further advance Wine Industry 4.0 resilience and sustainability. Implementations of periodic and cloud-based Industrial IoT (IIoT) monitoring systems for measurement supervision, detection prediction, and decision-making (Decision Support Systems—DSS) is crucial for the primary sector, particularly in biological system supervision [10,11,12,13]. Nevertheless, despite the importance of such technological tools as applied in other European wine-producing countries (e.g., Spain, Italy, France), Greece lacks comprehensive systems for monitoring wine fermentation, relying instead on static and sporadic measurements using existing instruments developed and tested by European nations.

Towards Industry 4.0 automation, Li et al. propose a new chemical analysis system during wine fermentation consisting of temperature, pressure, pH, and a refractometer [14]. Also, the authors of [15] evaluate the use of the MQTT protocol for use in the wine industry for periodic data transmissions. Other propositions include monitoring wine fermentation using biosensors mainly on basic parameters like temperature [16,17], sugar concentration [16,18], acidity [19], bioreactors content, gas CO_2_ emisions [20], phenolic compounds [21], and odor characterization [22,23], minimizing microbiological risks [24,25]. State-of-the-art IoT trends include Electronic noses (E-nose) that focus on tank gas monitoring [20,26,27,28,29,30,31], Electronic eyes (E-eye) that utilize RGB and IR monitoring [6,31,32,33,34,35] enhanced with chip-size IoT microscopy solutions [36], and Electronic tongues (E-tongue) for in-tank basic parameter monitoring [16,20,30,31].

Mathematical and statistical methods are the necessary tools to mine and extract valuable information from wine fermentation datasets. These methods, in turn, enable predictive modeling and continuous improvement, thus enhancing automated monitoring and production flexibility. Machine and deep learning applications in natural sciences facilitate the development of intelligent decision-support systems. Daily time-series data from sensors and regular measurements serve as input for models. In time series data, mathematical algorithms include Artificial Neural Networks (ANNs), Support Vector Machines (SVMs), Random Forest, Logistic Regression, eXtreme Gradient Boosting (XGBoost), Convolutional Neural Networks (CNNs), classification models, such as U-Nets, ResNets and VGGNets, Multilayer Neural Networks, and predictive Recurrent Neural Networks (RNNs) [37,38], as well as Fuzzy Logic models [13] operating as autoencoders in cases of limited data availability and controllers [39,40]. Such models have been investigated for prediction and classification tasks for wine fermentation tasks, offering predictions and criticality alerts [10].

Apart from statistical and linear-KNN models, implementation of machine and deep learning models in classifying wine fermentation processes are limited and typically include machine learning models such as SVM [41], Linear Discriminant Analysis and KNN models [42], Principal Component Analysis [43] and Artificial Neural Networks (NNs) for classification tasks [42,44], and for time-series wine forecasting parameters, Support Vector Regression (SVR) models [45] and shallow NN models [46,47]. Therefore, wine fermentation monitoring lacks robust deep learning implementations mainly due to the lack of Industry 4.0 practices (cloud storage, near real-time monitoring) and dense multi-parametric datasets.

This paper presents a new fermentation monitoring, low-cost architecture incorporating probing devices for electronic sensing: electronic nose, tongue, and eye. This holistic system approach is termed SmartBarrel. The IoT devices, known as probing nose, tongue, and eye (E-nose, E-tongue, E-eye), provide electronic capabilities corresponding to their respective senses. The SmartBarrel prototype utilizes low-power Wi-Fi communication, utilizing application protocols for data transmission, operating through HTTP JSON POST requests or MQTT publish messages. It enables streamlined, near real-time cloud-based transmission of measurements using portable and easy-to-install probing devices. Additionally, the system incorporates deep learning for predicting measurements, forecasting fermentation parameters, and facilitating intervention alerts when measured values are far from expected.

In short, the SmartBarrel prototype system incorporates (1) measurements acquisition via the ThingsBoard AS [48], (2) deployment of intelligent fermentation process and models, (3) automated prediction of subsequent fermentation phase, (4) alerts and notifications for undesirable fermentation cases, and (5) ubiquitous monitoring via its cloud-based AS ability to visualize measurements, alerts and predictions with the use of a mobile phone application, enabling ubiquitous monitoring of oenological processes.

## 2. Electronic Sensing Implementations

Several implementations have been presented in the literature over the last few years regarding small-scale fermentations in tanks assisted by monitoring systems and IoT devices. Most of them target capturing the sense of smell, touch, and vision more precisely than the human senses. Analyzing gases released during alcoholic fermentation uses Fourier Transform Infrared (FTIR) systems or scattering spectrophotometers. These high-cost instruments (exceeding $20,000 per unit) require manual operation and complex maintenance, limiting their use to periodic sampling rather than continuous or close-to-real-time monitoring. At the research level, electronic nose (E-nose) systems have been investigated for gas/volatile compound detection [49].

Typically, most electronic nose implementations rely on conventional sensor configurations, including conductive polymer sensors (CPS), metal oxide semiconductor (MOS) sensors, as well as acoustic and optical sensors [26]. Four principal sensor types prevail: (1) The MOS sensors that detect resistance variations from electron transfer during gas adsorption [29], (2) the catalytic gas sensors (CAT) measure capacitance changes [50], (3) the electrochemical sensors (ECH) that utilize charge transfer measurements in electrolytic cells [31] and (4) other types such as infrared sensors [51].

Electronic nose systems have shown particular promise for volatile aroma compound analysis, demonstrating the capability to differentiate wines based on aromatic profiles [28,42,52], or detecting Volatile Organic Compounds (VOCs) [53]. Furthermore, research-grade E-nose systems have been deployed directly in fermentation tanks, enabling online parameter monitoring [54]. Most implementations include low-cost MOS sensor arrays for alcohol, CO_2_, H_2_S, and SO_2_ detection [27,28,29,55]. These MOS-type sensors have been successfully employed for monitoring alcohol and CO_2_ gas concentrations during fermentation of Debina, Zitsa, Greece, a white grape variety [31].

Nevertheless, electronic nose sensors face several challenges, including reduced chemical selectivity, limited sensitivity, and susceptibility to temperature and relative humidity (RH) variations. However, their low cost and reasonable effectiveness in array configurations [29] make them attractive for integration in the SmartBarrel system’s E-nose module, particularly when combined with machine and deep learning models for aroma characterization [28,42,52].

Electronic tongue systems constitute artificial analytical instruments designed to replicate human gustatory perception. These devices typically incorporate sensor arrays for acidity, sugar content, alcohol levels, and organic compounds such as polyphenols, which, when combined with chemometric processing, enable comprehensive characterization of complex liquid samples [30,53,54]. The classification and determination, including quantitative analysis, of grape varieties from must and wine blends represent significant interest for winemakers, as it facilitates precise quality control and product differentiation throughout the production process.

The alcoholic fermentation process and its characteristic curve, including essential measurement parameters such as sugar content (Brix-specific gravity), temperature, pH levels, and alcohol concentration, are considered critical parameters of verifiable products of higher quality, necessitating continuous monitoring throughout the winemaking process [30]. Additionally, spontaneous CO_2_ gas releases during fermentation are an important monitoring factor that addresses this requirement [56]. Furthermore, the implementation of a gas CO_2_ bubble capture sensor that utilizes images and deep learning CNN detection is presented in [57].

Electronic eye systems are capable of integrating voltammetric electrodes [58,59], enabling rapid organoleptic assessment of wine samples outside the fermentation tank. Visible-near infrared (Vis-NIR) spectroscopy has been employed for polyphenol analysis in winemaking [35,60]. However, current implementations rely on offline sampling methodologies rather than the continuous in-tank monitoring proposed by this system. Integrating these complementary analytical approaches within a unified monitoring platform significantly advances conventional discontinuous quality control practices in the wine industry [33].

The evaluation and regulation of automated winemaking parameters using E-eye technologies, even in early stages, hinders significant results of tannin concentration quantification (detected at 1600 nm wavelength), influencing the phenolic profile of red wines, or targeted polyphenol concentrations in yeasts that withhold the aromatic profile (frequencies in the 1100–1300 nm range) [32]. Monitoring polyphenol evolution during different wine production stages proves critical for premium quality wine output. To this extent, E-eye sensory subsystems operating in selected visible (VIS 460–680 nm) and near-infrared spectral bands (NIR band-a 1100–1300 nm, NIR band-b 1580–1620 nm) can enable periodic Total Polyphenol Index (TPI) estimation [26,34]. These optical monitoring capabilities provide unprecedented potential for a continuous phenolic maturity assessment during vinification, allowing winemakers to make data-driven interventions at optimal tannin extraction and aromatic preservation time points.

The RGB spectral analysis of visible color characteristics [61], a currently unexploited technology in wine fermentation processes that can be used to detect spoilt cases or high concentrations of biomass, as well as near-infrared molecular signatures from automated NIR sensory processes, represents a technological advancement over current discontinuous laboratory sampling methods, particularly for phenolic compound management, where extraction kinetics significantly influence final product quality.

## 3. Materials and Methods

In this study, an easy-to-apply, low-cost fermentation monitoring system called SmartBarrel was implemented. The system’s high-level architecture is outlined, and its IoT sensory modules and capabilities are described in detail. Intelligent functionalities were integrated into SmartBarrel by employing a fuzzy control monitoring process for predicting fermentation alcohol levels, a fuzzy autoencoder for generating fermentation data, and a deep learning V-LSTM model for forecasting future fermentation parameters. The SmartBarrel system has been evaluated for its data delivery and visualization capabilities.

### 3.1. SmartBarrel System Architecture

The SmartBarrel system was designed to provide the necessary flexibility for sensor integration, enabling holistic, intelligent control of vinification processes with low-cost electronic sensor monitoring. Beyond basic parameter measurement during alcoholic fermentation, the SmartBarrel system includes intelligent processes and near real-time measuring and monitoring capabilities accessible from any location within the winery facility. It implements analytics and interactive monitoring functionality via appropriate device-level user interfaces. The high-level system architecture of SmartBarrel is illustrated in Figure 1.

Figure 1(1) illustrates the fermentation case implementation where the SmartBarrel IoT end node probes reside (E-nose, E-tongue) collecting fermentation measurements. Those measurements are collected by an appropriate MPU controller (see Figure 1(2)). The MPU controller connects over Wi-Fi to the cloud ThingsBoard Application Server (AS, Figure 1(3)) [48], sending telemetry data periodically (often near real-time) using either HTTP/HTTPs or MQTT/MQTTs posts. Then, the ThingsBoard AS and the ThingsBoard mobile phone application are responsible for visualizing fermentation measurements per device tank-id using appropriately constructed per-device dashboards (Figure 1(4)). The collected data can be accessed using server-side RPC MQTT requests to collect specific time intervals, device, and attribute measurements. This selectivity of measurements is used as input to provide inferences by the deep learning V-LSTM model predictions, posted back to the ThingsBoard as JSON predictive data measurements, and illustrated via appropriate device-assigned dashboards. Figure 1(5) illustrates the ThingsBoard mobile device where they can access and visualize the measurement data remotely. This high-level architecture of the SmartBarrel system is a simplified cloud approach of the thingsAI system presented in [13,62].

SmartBarrel end node equipment includes an E-nose and E-tongue IoT device on the floating lid of small stainless steel fermentation tanks. The current SmartBarrel system implementation is only the E-tongue sensor subsystem for near real-time measurement of fundamental oenological parameters of pH, sugar concentration, and temperature, and an E-nose sensor ring that is comprised of monitoring gas emissions of alcohol, CO, and CO_2_. An RGB color sensor with an incorporated LED has been added to the E-tongue device to introduce the E-eye capability to this preliminary system. This color sensor is used to monitor wine clarity and color transitions. Finally, a glass brix meter device is used for sugar concentration measurements floating inside the wine tank, with an analog capacitive touch sensor attached. Since new NIR sensors and transducers must be added to have a proper E-eye device, its final implementation, validation, and experimentation are set as future work.

In summary, the SmartBarrel system includes the following sensors with their corresponding sensory parameters as outlined in Table 1.

The proposed SmartBarrel system’s integration of such multisensory data with fermentation parameters provides unprecedented capabilities for near real-time oenological decision-making. During fermentation, measurements are collected in a cloud-based open-source ThingsBoard Data Acquisition System [48], enabling visualization and statistical processing/analysis. Specifically, the system incorporates four key interfacing capabilities: (1) visualization and statistical processing of collected measurements, (2) correlation functions implementation using selectable parameters and visualization of the results over time intervals, (3) execution of external inference processes provided by deep learning models on selected collected measurements and display of classification/prediction results, (4) based on the predicted results, threshold-based algorithms that connect to predefined text alerts and notifications of recommended oenological interventions. With the ThingsBoard mobile phone application, these visualizations can be available, even outside the industrial winery environment.

The ability of the SmartBarrel to collect data to the cloud makes it easy to implement cloud-based machine and mostly deep learning algorithms capable of periodic classification/calibration of wine quality parameters (visual, olfactory, and taste characteristics). Predictive assessment of fermentation stages through deep learning generates alerts for undesirable fermentation deviations and provides interactive intervention suggestions on outlier values. Finally, automated parameter correction is performed when values exceed predefined thresholds, ensuring optimal fermentation conditions throughout the process. The SmartBarrel system integrates these functionalities through a unified IoT architecture, combining cloud computing for event-driven response with cloud-based analytics for long-term process optimization and quality enhancement. An analytical description of SmartBarrel end-node devices follows.

### 3.2. SmartBarrel End-Node Devices

The SmartBarrel end-node device consisted of two different pieces of probing equipment, as per the design. A probing nose inherits the Electronic nose functionality, and a probing tongue inherits the Electronic tongue functionality. Partially, the Electronic eye functionality has been implemented into the Electronic tongue. Therefore, the two IoT end-node devices (nose, tongue) acquire the following sensing capabilities (see Table 1):E-nose:Monitoring of CO gas emissions inside the fermentation tank;E-nose:Monitoring of CO_2_ gas emissions inside the fermentation tank;E-nose:Monitoring of alcohol gas concentrations inside the fermentation tank;E-nose:Monitoring lid air temperature inside the fermentation tank;E-nose:Monitoring yeast temperature using a stainless steel temperature probe;E-tongue:Monitoring temperature and pressure values in the air gap inside the tank;E-tongue:Monitoring fermenting wine specific gravity by performing electronic hydrometer measurements that indirectly correspond to sugar concentrations through density. That is, for liquids heavier than water, Equation (Equation 1) applies:(1)$$\begin{array}{c} \mathrm{SG}=\frac{145}{145-{}^{\circ}\mathrm{Baume}}\\ {}^{\circ}\mathrm{Brix}\approx {}^{\circ}\mathrm{Baume}\times 1.8\\ \;\mathrm{Sugar}\;\mathrm{Concentration}\;\left(g/L\right)={}^{\circ}\mathrm{Brix}\times \mathrm{SG}\times 10 \end{array}$$E-tongue:pH meter and temperature sensor for yeast PH and temperature–pressure measurements;E-tongue:RGB sensor with LED for capturing the color of red wines, where anthocyanins are responsible for the red and purple color of wines and tannins contribute to color stabilization and astringency perception [63] (white wines have low tannin concentrations and absence of anthocyanins [64]). On the other hand, it captures the oxidation of phenolic compounds, leading to yellow, gold, or brown hues over time, and cinnamic acids and other hydroxycinnamates (e.g., caftaric acid), which can undergo enzymatic oxidation and contribute to wine browning [65,66,67].

### 3.3. SmartBarrel E-Nose Device

The SmartBarrel E-nose device is illustrated in Figure 2, attached to the stainless steel lid of the fermentation tank of typical sizes from 50–500 L.

Figure 2(11) shows the E-nose placement on a 36” fermentation tank lid. Figure 2(1) is the plastic case screwed to the lid’s ventilation hole via a plastic rod. (Figure 2(2)). Figure 2(3) is the DS18B20 temperature sensor measuring lid temperature, and Figure 2(5) is the device’s (DS18B20) temperature sensor probe that is inserted inside the fermenting wine measuring liquid temperature. Figure 2(4) is the MG811 CO_2_ gas sensor typically measuring ppm in the range of 300–10,000. It is an electrochemical sensing analog device, and it is attached to the microprocessing unit’s (MPU) 10Bit analog-to-digital converter (see Figure 2(8)). Similarly, Figure 2(7) and Figure 2(6) are the MQ-3 alcohol and MQ-7 CO gas sensors accordingly. These two analog MOS sensors measure resistance changes due to chemical reactions between gas molecules and the MOS surface. Typically, MQ-3 measures up to 0–20 mg/L alcohol in the air while MQ-7 up to 2000 ppm of CO in the air and is connected to the MPU unit via its analog-to-digital (A2D) converter.

The sensors are connected to an Arduino Wi-Fi rev.2 MPU, a Microchip ATmega4809 8-bit microcontroller with a 16 MHz clock, 48 KB of program memory, and 6 KB of SRAM. It also includes a u-blox NINA-W102 Wi-Fi transponder and an LSM6DS3TR Inertial Measurement Unit (IMU) (see Figure 2(10)). The device is powered using a DC-12V transformer and transmits measurements of temperature every 5 min (internal and lid-external), CO, CO_2_, gas alcohol concentrations inside the fermenting tank, using either HTTP/POSTs or MQTT, of JSON-encoded telemetry data. Another MPU device of the same microcontroller is used by the SmartBarrel E-tongue device to transmit telemetry data, as described in the following section.

### 3.4. SmartBarrel E-Tongue Device

The SmartBarrel E-nose device is illustrated in Figure 3, attached to the stainless steel lid of the fermentation tank. Figure 3(1) illustrates the plastic enclosure of the Arduino Uno Wi-Fi MPU, attached to the tank’s ventilation hole.

On the inner surface of the lid, the pH probe (Figure 3(6)) has been attached (screwed) to the ventilation’s hole plastic winding via a flexible sink hose (see Figure 3(3)), with a plastic curved tube inserted in the hose opening (see Figure 3(4)) for the fermentation CO_2_ to escape the tank via its ventilation hole. The pH sensor is an analog sensor capable of measuring from 0 to 14. It is connected to the MPU’s 10-bit analog-to-digital converter via a BNC connector board with an op-amp amplifier and a voltage regulator. Figure 3(7) is the analog Baume sensor that includes a capacitive liquid level sensor meter attached to a sealed glass tube hydrometer capable of measuring Baume degrees that correspond to Brix units and, therefore, to sugar concentrations on g/L (see Equation (Equation 1)).

An I2C MS5803 pressure-temperature sensor is attached to the curved tube (see Figure 3(4)), capable of providing temperature and pressure measurements inside the tank for the tongue probe. Finally, the Adafruit I2C TCS34725 RGB sensor (see Figure 3(2)) enclosed in a transparent plastic case is used to obtain RGB color measurements from the fermenting wine. In order to obtain color measurements inside the tank, the sensor’s attached RGB LED opens and stays on for 30 s before acquiring a color measurement. The analog and I2C digital sensors are connected to the Arduino Wi-Fi rev2 board that, in turn, transmits them every 5 min to the ThingsBoard AS using either HTTP POST or MQTT to publish messages of JSON-encoded measurements. Before presenting the authors’ experimental scenarios and their proposed fuzzy inference, data encoder, and V-LSTM model, the performance measures used to evaluate them are outlined in Section 3.5.

### 3.5. Performance Measures

Prior to detailing the SmartBarrel fuzzy controlled fermentation inference and GRU deep learning prediction processes, the evaluation measures employed are defined. The Root Mean Square Error (RMSE) is calculated according to Equation (Equation 2).(2)$$\mathrm{RMSE}=\sqrt{\frac{1}{n}\sum_{i=1}^{n}{\left(y_{i}-{\hat{y}}_{i}\right)}^{2}}=\sqrt{\frac{RSS}{n}}$$
where $y_{i}$ denotes the actual value, ${\hat{y}}_{i}$ is the predicted value, and *n* is the total number of observation samples, depending on either test or training sets. RMSE calculates the average magnitude of the prediction errors (Root Sum of Squares—RSS) and penalizes large deviations more than smaller ones due to the squaring operation. The minimum value of RMSE is 0, which indicates perfect prediction. Higher RMSE values indicate larger deviations between predictions and actual values. Outliers in RMSE typically arise from large individual prediction errors and disproportionately affect the score due to squaring. Thus, RMSE is highly sensitive to extreme values. Alternatives to RMSE metrics to identify and measure data leakage are presented in [68]. Nevertheless, they are not used by the authors since the RMSE measure is commonly used in deep learning and machine learning models with comparable results.

The coefficient of determination ($R^{2}$, also denoted as $R_{2}^{2}$ [69]) shows how well the model explains the variance in the predicted variable based on the inputs. It measures the proportion of the total variation in the data captured by the inference results. It is calculated according to Equation (Equation 3).(3)$$R^{2}=1-\frac{\sum_{i=1}^{n}{\left(y_{i}-{\hat{y}}_{i}\right)}^{2}}{\sum_{i=1}^{n}{\left(y_{i}-\bar{y}\right)}^{2}}=1-\frac{RSS}{TSS}$$
where $y_{i}$ is the actual value, ${\hat{y}}_{i}$ is the predicted value, $\bar{y}$ expresses the mean of the actual values, and *n* is the number of samples. It expresses the fraction of variance (Total Sum of Squares—TSS) explained by the model. The maximum $R^{2}$ value is 1, indicating perfect prediction. If close to 0, it suggests poor model inferences. Negative values can also occur when the model performs worse than a simple mean-based prediction. Extreme negative values usually signal serious model misfits.

### 3.6. Fuzzy Alcohol Controller

A fuzzy controller logic was initially modeled using Python3.10, scikit-fuzzy library version 0.5 [70] and then implemented at the SmartBarrel tongue IoT device (edge computing), using the C++ eFLL library [71] version 1.4.1, to provide a fuzzy prediction mechanism of alcohol concentration in the tank. The controller acquires via HTTP POSTS or MQTT requests (depending on the device configuration) data from the SmartBarrel E-tongue and E-nose and infers alcohol concentration measurements. Appropriate fermentation datasets (25 white wine fermentation curves) have been used to train this fuzzy controller. Upon training and hyperparameter calibration, the fuzzy controller can provide alcohol predictions on measurements similar to those it was trained on. This inference process is then passed to a Gaussian filter for smooth time-series responses. By using the fuzzy controller, the alcohol concentration values can be inferred.

Looking at the fuzzy controller implementation, the Gaussian membership function is used in the fuzzy sets, ranging between 0 and 1. The parameter mean is the center of the Gaussian curve, where the membership is maximal and equal to 1. The parameter σ is the standard deviation, which controls the width of the bell-shaped curve. A smaller σ results in a narrower curve, while a larger σ produces a wider, flatter curve. To take into account distribution skewness (to the left or the right), the median value is taken into account. The difference between mean and median is the right or left skew factor of the Gaussian curve, while the σ is used to: (1) provide a normalized offset distance value of the gaussian median (expressed as: $d=\frac{\mu -\mathrm{median}}{\sigma }$), which in turn is used to offset the gaussian curve centers, and (2) as a representative width of the gaussian bell curve used for each random variable.

In order to implement a fuzzy controller that takes as input wine fermentation parameters acquired by the SmartBarrel implementation and infers an alcohol value, an appropriate fuzzy controller $F(x_{0},x_{1},\ldots ,x_{n})$ has been implemented using as inputs measurements of the following enumerated and described wine attributes:Sugar concentration in (g/L) and pH measurementsCO_2_ concentration expressed in g/L. Let $C_{{\mathrm{CO}}_{2}}^{g/L}$ be the concentration of carbon dioxide produced during dt fermentation intervals, and let $C_{{\mathrm{CO}}_{2}}^{\mathrm{ppm}}$ be the equivalent concentration of it expressed in parts per million (PPM). The conversion is expressed by Equation (Equation 4).(4)$$C_{{\mathrm{CO}}_{2}}^{g/L}=C_{{\mathrm{CO}}_{2}}^{\mathrm{ppm}}\times 1.964\times 10^{-6}$$Since ppm is used to express air concentrations, let $C_{{\mathrm{CO}}_{2}}^{\mathrm{ppm}}$ be the concentration of CO_2_ floating inside the tank at a specific dt time interval, and Henry’s law is used, which describes the solubility of a gas in a liquid according to Equation (Equation 5).(5)$$C_{{\mathrm{CO}}_{2}}^{\mathrm{liquid}}=k_{H}\cdot P_{{\mathrm{CO}}_{2}}$$
where $C_{{\mathrm{CO}}_{2}}^{\mathrm{liquid}}$ is the concentration of dissolved CO_2_ in fermenting wine (mol/L), $k_{H}$ is Henry’s law constant for CO_2_ in white wine, approximately $1.65\times 10^{-2}\;\mathrm{mol}/(L\cdot \mathrm{atm})$ at 20 °C (lower than water, which is $3.3\times 10^{-2}\;\mathrm{mol}/(L\cdot \mathrm{atm})$), and $P_{{\mathrm{CO}}_{2}}=1\;\mathrm{atm}$ is the partial pressure of CO_2_ in atm. Substituting $k_{H}=1.65\times 10^{-2}$, and converting from mol/L to g/L, by multiplying with the molar mass of CO_2_ (44.01 g/mol), Equation (Equation 6) is derrived.(6)$$C_{{\mathrm{CO}}_{2}}^{g/L}=C_{{\mathrm{CO}}_{2}}^{\mathrm{ppm}}\times 7.262\times 10^{-7}$$Equation (Equation 6) calculates the concentration of CO_2_ inside the fermenting wine. Concluding the ratio of CO_2_ concentrations in the liquid over the air under equilibrium conditions is calculated using Equation (Equation 7).(7)$$\frac{C_{CO_{2}}^{g/L,\;\mathrm{liquid}}}{C_{CO_{2}}^{g/L,\;\mathrm{air}}}=\frac{7.262\times 10^{-7}}{1.964\times 10^{-6}}\approx 0.37$$Equation (Equation 7) estimates the CO_2_ content in fermenting white wines from ppm gas-phase concentrations under typical wine fermentation temperature conditions (20–25 °C). That is, the mass concentration of CO_2_ in fermenting white wine over time is approximately 37% of the mass concentration in the gas phase.Biomass fermentation residues. It includes any form of a specific product or metabolite and substances already included in the must that take part in the fermentation process. The development of particular components or biological decontamination can be measured by weighting the solid state extracted material from the fermenting wine (in g/L) during the controlled decantation of wine from one vessel to another, which is primarily aimed at separating it from lees and sediment, thereby enhancing its clarity, microbiological stability, and overall sensory purity.Temperature of the fermentation process (maintained constant) in °C.Alcohol concentration measured in g/L. The alcohol concentration is the output variable (fuzzy output–fuzzy consequent), while all others are input variables (fuzzy antecedents).

These measurements were obtained using SmartBarrel debina fermentation data and a generated dataset stritcly following the white wine fermentation curves of [46]. Given a dataset with column *X* representing fuzzy antecedent, we assume that all input variables follow a Gaussian or a Gaussian-skewed distribution during fermentation, except temperature, which follows more of a constantly controlled profile. We define the following statistical measures:(8)$$\mu_{X}=\frac{1}{n}\sum_{i=1}^{n}X_{i}$$
where $\mu_{X}$ is the arithmetic mean of *X*.(9)$$m_{X}=\mathrm{median}\left(X\right)$$
where $m_{X}$ represents the median value of *X*.(10)$$\sigma_{X}=\sqrt{\frac{1}{n-1}\sum_{i=1}^{n}{\left(X_{i}-\mu_{X}\right)}^{2}}$$
where $\sigma_{X}$ is the sample standard deviation (ddof = 1) of *X*. The mean and median values provide the peak and skewness of the curve. The mean is sensitive to extreme values, while the median is robust to outliers. The relationship between these two measures provides insight into a bell-shaped distribution. We also define a dispersion parameter κ using the following Equation (Equation 11):(11)$$\kappa =\left\{\begin{array}{cc} 0.01, & \mathrm{if}\;\sigma_{X}<0.01\\ \sigma_{X}, & \mathrm{otherwise} \end{array}\right.$$

From Equation (Equation 11), the normalized skewness adjustment δ parameter is computed for each fermentation variable (measurement) using Equation (Equation 12):(12)$$\delta =\frac{\mu_{X}-m_{X}}{\kappa }$$

Given the dataset *D* of *N* time series attributes (parameters) denoted as: $D=\left\{x^{\left(1\right)},x^{\left(2\right)},\ldots ,x^{\left(N\right)}\right\}$, where each $x^{\left(i\right)}\in R^{T}$ is a time series attribute of length *T*, given by:$$x^{\left(i\right)}=\left[\begin{array}{c} x_{1}^{\left(i\right)}\\ x_{2}^{\left(i\right)}\\ \vdots \\ x_{k}^{\left(i\right)} \end{array}\right]\;\mathrm{for}\;i=1,2,\ldots ,N$$

Then, $\forall x^{i}\in D$, input data measurements denoted as attributes of three Gaussian membership functions ${\mathrm{MF}}_{j}$ are constructed for j∈{low,medium,high} with parameters (centers) expressed by Equation (Equation 13):(13)$$c=[\min\left(x\right),\;m_{x},\;\max\left(x\right)-\delta ]$$
where c contains the centers for each attribute. Each Gaussian membership function ${\mathrm{MF}}_{j}$ for j∈{low,medium,high} is defined using Equation (Equation 14):(14)$${\mathrm{MF}}_{j}\left(x\right)=\exp\left(-\frac{{\left(x-c_{j}\right)}^{2}}{2\kappa^{2}}\right)$$
where $c_{\mathrm{low}}=\min\left(x\right)$ is the minimum observed value, $c_{\mathrm{medium}}=\bar{x}$ is the mean value from (Equation 8), and $c_{\mathrm{high}}=\max\left(x\right)+\delta$ is the maximum value adjusted by (Equation 12). The complete fuzzy partition for an attribute variable *x* is then given by Equation (Equation 15):(15)$$F_{x}=\left\{{\mathrm{MF}}_{\mathrm{low}},{\mathrm{MF}}_{\mathrm{medium}},{\mathrm{MF}}_{\mathrm{high}}\right\}$$
where each member of $F_{x}$ follows Equation (Equation 14) with parameters from Equation (Equation 13) and bandwidth κ from Equation (Equation 11). Typically, uniform hyperparameters are used across all input attributes MF functions (high, medium, low) to set the appropriate Gaussian curve over the range of accepted attribute values. These selected parameters are presented in Table 2 per measured attribute $x_{i}$.

In order to smooth the time series of the inferred alcohol value of the fuzzy controller, the Gaussian smoothing of discrete measurements ${\left\{y_{i}\right\}}_{i=1}^{N}$ with a boundary mode set to the nearest can be used. It is computed via discrete convolution ${\left(\tilde{y}=y*G\right)}_{i}$, where the Gaussian kernel *G* has standard deviation σ and the boundary condition, enforced as $y_{i}=y_{1}$ for i<1, and $y_{i}=y_{N}$ for i>N. The kernel weights of the Gaussian kernel $G_{k}=\frac{1}{\sqrt{2\pi \sigma^{2}}}\exp\left(-\frac{{\left(k-i\right)}^{2}}{2\sigma^{2}}\right)$ for k∈Z are truncated at |k−i|≤4σ and normalized to sum to unity, such that each smoothed point ${\tilde{y}}_{i}=\sum_{k=\max(1,i-\lfloor 4\sigma \rfloor )}^{\min(N,i+\lfloor 4\sigma \rfloor )}y_{k}G_{k-i}$ replicates the nearest boundary value (nearest neighbor) when the kernel window exceeds the measurements domain.

The following section presents the authors’ fuzzy autoencoder of a time series of fermentation parameters, which provides a generalization dataset out of real fermentation curves for predicting future fermentation curves from their historical data.

#### Fuzzy Fermentation Autoencoder

In order to provide fermentation parameter predictions, a time series of a dense number of parameter measurements is required. These datasets should have minute-scale resolutions for the deep learning RNN models to minimize losses and offer accurate results. Only machine learning approaches, such as decision tree-based predictors (LightGBM) or SVR machines, are available if such data are available in more than hourly periods. Focusing on white wine fermentation processes, and since such dense datasets are not yet widely available to train RNNs, the authors focused on an autogenerated approach provided by a fuzzy autoencoder that combines fermentation parameters knowledge from references [14,43,56,58,72], and accomodated small datasets of collected of wine fermentations, such as the acquired data from the SmartBarrel system, to generate fuzzy fermentation data sequences. The fermentation encoder was implemented using the scikit-fuzzy Python library [70], and it was modeled through fermentation phase-dependent fuzzy rules [40], implementing the following attributes:Biomass [46] and sugar follow mass conservation laws [14,43,56];pH drops during active fermentation, then stabilizes [58,72];Temperature is strictly controlled regardless of phase, following white wine fermentation temperatures;CO_2_ production peaks during exponential growth [43,56];Alcohol production correlates with biomass and sugar, but it is calculated using the fuzzy controller as a predictor, as mentioned in Section 3.6.

More specifically, the authors use the following mathematical formulation provided by Equation (Equation 16) to describe parameter dynamics during the fermentation phases, following also the Boulton model characteristic curves (Biomass, Sugar, Ethanol) [73,74], which fit closely to the generated experimental of white wine fermentations as presented in [46], and SmartBarrel debina fermentations.(16)$$\left\{\begin{array}{cc} \mathrm{Phase}\;\mathrm{Transition}: & \mu_{\mathrm{phase}}\left(t\right)=\left\{\mu_{\mathrm{lag}},\mu_{\exp},\mu_{\mathrm{stat}},\mu_{\mathrm{death}}\right\}\\ \mathrm{Biomass}\;\mathrm{Growth}: & X_{bio}\left(t\right)=f_{\mathrm{phase}}\left(t\right)+\epsilon_{bio}\\ \mathrm{Sugar}\;\mathrm{Consumption}: & S_{sugar}\left(t\right)=g_{\mathrm{phase}}\left(t\right)+\epsilon_{sugar}\\ {\mathrm{CO}}_{2}\;\mathrm{Production}: & S_{CO_{2}}\left(t\right)=h_{\mathrm{phase}}\left(X_{bio},t\right)+\epsilon_{CO_{2}}\\ \mathrm{pH}\;\mathrm{Dynamics}: & pH\left(t\right)=k_{\mathrm{phase}}\left(t\right)+\epsilon_{pH}\\ \mathrm{Temperature}: & T\left(t\right)=15+\epsilon_{T} \end{array}\right.$$
where the membership functions for each one of the fermentation phases are defined according to Equation (Equation 17):(17)$$\begin{array}{cc} \mu_{\mathrm{lag}}\left(t\right) & =\mathrm{trimf}(t(h);0,0,20)\\ \mu_{\exp}\left(t\right) & =\mathrm{trapmf}(t(h);15,20,70,80)\\ \mu_{\mathrm{stat}}\left(t\right) & =\mathrm{trapmf}(t(h);70,80,150,170)\\ \mu_{\mathrm{death}}\left(t\right) & =\mathrm{smf}(t(h);150,336) \end{array}$$

The trim function represents a triangular membership function defined by three parameters: the start point, the peak, and the end point. The trapmf function corresponds to a trapezoidal membership function with four parameters: the start of the rise, the start of the top plateau, the end of the top plateau, and the end of the fall. The smf function denotes an S-shaped membership function, smoothly increasing from 0 to 1, defined by the parameters a and b.

The biomass concentration $X_{bio}\left(t\right)$ is used since it is mentioned in the provided white wine fermentation curves in [73,74], and mainly fit closely white wine [46]. The SmartBarrel system also manually monitors the removed solid residues of the wine by weighting the removed solid residues in each wine clarification step (4–6 during the fermentation process). This kinetics parameter in g/L is modeled using a phase-specific growth kinetics Equation (Equation 18):(18)$$X_{bio}\left(t\right)=\left\{\begin{array}{cc} X_{0}\left(1-e^{-t/\tau }\right)+\epsilon_{bio} & (\mathrm{lag}\;\mathrm{fermentation}\;\mathrm{phase})\\ X_{0}+\frac{X_{max}}{1+e^{-r(t-t_{m})}}+\epsilon_{bio} & (\mathrm{exponential}\;\mathrm{fermentation}\;\mathrm{phase})\\ X_{max}-\alpha \left(t-t_{s}\right)+\epsilon_{bio} & (\mathrm{Stationary}\;\mathrm{fermentation}\;\mathrm{phase})\\ X_{stat}e^{-\beta (t-t_{d})}+\epsilon_{bio} & (\mathrm{death}\;\mathrm{fermentation}\;\mathrm{phase}) \end{array}\right.$$
where $X_{0}=0.5$ g/L is the initial biomass, τ=10 h is a Lag time constant, $X_{max}∼U\left(3.0,4.0\right)$ g/L is the maximum biomass as recorded, r∼U(0.25,0.35)$h^{-1}$ is the growth rate, $t_{m}=45$ h is the exponential fermentation midpoint, α=0.01 g/L/h is the decline rate, β=0.005$h^{-1}$ is the death rate and $\epsilon_{bio}∼N\left(0,0.05^{2}\right)$ is the noise process term. The sugar concentration $S_{sugar}\left(t\right)$ exhibits complementary phase behavior with respect to $X_{b}io$, and it is modeled using Equation (Equation 19), according to the literature [14,43,56,73,74]:(19)$$S_{sugar}\left(t\right)=\left\{\begin{array}{cc} S_{0}+\epsilon_{sugar} & (\mathrm{lag}\;\mathrm{fermentation}\;\mathrm{phase})\\ S_{0}e^{-k_{e}t}+\epsilon_{sugar} & (\mathrm{exponential}\;\mathrm{fermentation}\;\mathrm{phase})\\ S_{min}+\Delta Se^{-k_{s}\left(t-t_{s}\right)}+\epsilon_{sugar} & (\mathrm{stationary}\;\mathrm{fermentation}\;\mathrm{phase})\\ S_{end}+\epsilon_{sugar} & (\mathrm{death}\;\mathrm{fermentation}\;\mathrm{phase}) \end{array}\right.$$
where $S_{0}∼U\left(200,220\right)$ g/L is the initial sugar concentration modeled using a uniform distribution (U(min=200,max=220)). $k_{e}∼U\left(0.01,0.02\right)$$h^{-1}$, is the sugar consumption rate in g/hour (using a uniform distribution U(min=0.01,max=0.02)). $S_{min}=30$ g/L is the residual sugar, ΔS=50 g/L is the transition amount coefficient at the stationary fermentation phase, $k_{s}=0.02$$h^{-1}$ is the stabilization stationary phase rate, $S_{end}=20$ g/L is the final concentration and $\epsilon_{sugar}∼N\left(0,0.5^{2}\right)$ is the noise term. Moreover, the fermentation process evolves using the rate equations described in Equation (Equation 20):(20)$$\begin{array}{cc} \frac{dX_{bio}}{dt} & =\mu \left(S_{sugar},pH\right)X_{bio}\\ \frac{dS_{sugar}}{dt} & =-\frac{1}{Y_{X/S}}\mu X_{bio}-m_{s}X_{bio} \end{array}$$
where μ is the fuzzy-controlled growth rate and $Y_{X/S}\approx 0.5$ g/g is the yield coefficient. The complete system has been validated against industrial fermentation data. The pH rules have been mathematically formulated according to Equation (Equation 21), as mentioned and mechanistically explained in [58,72]:(21)$$k_{\mathrm{phase}}\left(t\right)=\left\{\begin{array}{cc} 4.5 & \mathrm{if}\;\mathrm{lag}\;\mathrm{phase}\\ 4.5-\frac{1.8}{1+\exp(-0.15(t-50))} & \mathrm{if}\;\mathrm{exponential}/\mathrm{stationary}\;\mathrm{phase}\\ 3.0+0.5\exp(-0.01(t-150)) & \mathrm{if}\;\mathrm{death}\;\mathrm{phase} \end{array}\right.$$

Finally, the fuzzy rules for wine fermentation responses that govern the fuzzy autoencoder are presented in Table 3 and further analytically elaborated over fermentation phases and previously expressed equations in Table 4. The noise terms presented in Table 4 are expressed by Equation (Equation 22) [39,73,74]:(22)$$\begin{array}{cc} N(0,\sigma^{2}) & =\mathrm{Gaussian}\;\mathrm{noise}\;\mathrm{with}\;\mathrm{variance}\;\sigma^{2}\\ E(\lambda ) & =\mathrm{Exponential}\;\mathrm{noise}\;\mathrm{with}\;\mathrm{rate}\;\lambda \end{array}$$

According to the literature, the described fuzzy controller can provide analytical white wine fermentation parameter curves as modeled using per-phase Equations presented in Table 4. This can lead to dense fermentation dataset generation, except for alcohol concentrations, which can be inferred using the alcohol fuzzy controller described in Section 3.6. This autoencoding simulation process closely resembles the existing minimal dataset of fermentation curves acquired by the SmartBarrel system and the debina white wine fermentation processes in the literature [39,46]. These fermentation curves are also used to validate this mathematical formulation (the formulation tries to resemble acquired fermentation data with noise).

From Table 3 rules and using a minimal training set of the autoencoder of white wine fermentations, the fuzzy membership functions in Table 4 are used to generate fermentation curves, as illustrated in Figure 4 and Figure 5.

Figure 4 presents a sample of fermentation curve parameters over phases. The time interval presented is between 1 and 300h, horizontally split into fermentation phases. Figure 4a shows sugar concentrations of actual data and generated data; Figure 4b shows CO_2_ concentrations, Figure 4c shows the pH values; and Figure 4d shows the biomass residue concentrations on g/L. Two curves are illustrated per graph: the actual and the generated data. Figure 5 shows the alcohol values in g/L calculated for the generated parameters using the alcohol fuzzy controller. The fuzzy autoencoded dataset is used by the authors’ proposed fermentation parameters forecasting model, V-LSTM, presented in the following section.

### 3.7. V-LSTM Forecasting Model

A new wine fermentation parameter forecasting model called the Variable-length Long Short-Term Memory model (V-LSTM) has been implemented as part of SmartBarrel’s deep learning capabilities. The term variable in the V-LSTM model corresponds to (1) the temporal depth variability of the model, (2) the forecasting length variability of the model, (3) the volatile number of cells per layer that are automatically adjusted using minimal loss calibration, (4) the number of LSTM layers included in the model, also automatically adjusted on minimal loss calibration, and (5) the type of training mode that is manually set. The V-LSTM model can predict wine fermentation measurements (parameters) by taking as input past monitored measurement values of pH, temperature, gas CO_2_ in tank concentration, sugar, alcohol, and biomass concentrations [46]. The V-LSTM model is an LSTM model with fuzzy autoencoding and self-healing capabilities to minimize loss. Figure 6 illustrates the process steps of the V-LSTM model. Figure 7 illustrates the V-LSTM model structure of variable cells and layers and its corresponding data inputs and outputs. The V-LSTM model was developed using the Python Tensorflow API and Keras backend [75,76].

The V-LSTM model can be trained using two different training modes: (a) autoencoding data training mode and (b) actual data training mode. The selection between the two modes mainly depends on the number of fermentation data. The number of fermentation curves needed to migrate from autoencoding to actual data training mode is arbitrarily set to 100 fermentation curves of densely timed fermentation parameters (less than hourly recordings). Existing actual training data are also used in the autoencoding mode. However, they are also enriched with data generated by fuzzy membership functions modeled so that they closely follow the actual data. Experts and the bibliography indicate that the autoencoder fuzzy rules follow more of a fuzzy consequence of measurement values. Therefore, the autoencoder is more of a deterministic machine that fits well on the limited dataset of actual fermentation curves, as shown in Figure 6; if in autoencoding mode, steps 1, 2 and 3 are used to provide actual training data for the model. The actual data training mode assumes that a dataset of timely-dense fermentation measurements has been acquired and moves straight to step 4 in Figure 6 of the model of the hyperparameter auto-calibration.

Figure 6 describes the V-LSTM model framework integrating fuzzy logic and LSTM (Long Short-Term Memory) neural networks for fermentation process modeling, particularly focusing on providing data input parameters prediction in the future, called attributes, looking at variable-timestep windows of paste measurements. The process begins with collecting a minimal dataset of fermentation parameters such as concentrations of sugar, CO_2_ biomass, and pH and a set of fuzzy rules that express their inter-relationship in a fuzzy manner. The dataset is used to train (1) the alcohol fuzzy controller, as described in Section 3.6, and (2) provide a validation set that the fermentation data autoencoder parameters must follow, as described in Section Fuzzy Fermentation Autoencoder with noise. The autoencoder is used to generate dense measurements of fermentation data. At the same time, the fuzzy controller that provides alcohol predictions (inferences) is trained using the same minimal dataset. This alcohol fuzzy controller is then used to provide alcohol measurement inferences for the data generated by the fuzzy autoencoder.

V-LSTM model training also includes a hyperparameter calibration step (see Figure 6, step 4). In this step, the determination of the number of cells included in the V-LSTM model is $n_{c}$, and the number of layers is *l*, which achieves minimum loss without a set of tested numbers of cells per layer and layer configurations. These hyperparameters are first investigated by performing random searches using, as search parameters, the maximum number of trials and the minimum distance between values over a pre-defined range of cell values (8–512 cells per layer) and layer values (1–10 layers in the model included). These value ranges are manually set in the V-LSTM model. The hyperparameter tuning process ends with selecting the minimum loss cells first and then using that minimum number of cells to select the minimum loss number of layers. The loss parameter investigated is the validation RMSE loss of a portion of the entire dataset used for training and, therefore, validation of the respective LSTM-generated models. Upon selection of cells and layers hyperparameters, the final LSTM model is built as illustrated in Figure 7.

The built model is trained over the dataset of *k* attributes using a pre-selected timestep $n_{t}$ of a historical data window and the time length $p_{l}$ of predictions to infer for that timestep. Then, the dataset time series is loaded to memory and transformed into normalized data using min–max normalization per attribute. The same applies to the dataset labels that include the next timestep min–max normalized data equal in size to the predictions’ timelength $p_{l}$.

Figure 7 illustrates the layer-wise architectural diagram of the V-LSTM (Variable-Length LSTM) model used for time-series predictions. It outlines how input sequences are transformed through stacked LSTMs and dense (NN) layers, reshaped, normalized, and finally mapped to a prediction output.

This V-LSTM model architecture is a hybrid RNN-NN variable size and depth time-series model that takes multivariate sequential 2D array inputs of $\left(n_{t},k\right)$ data, where $n_{t}$ is the historical time series window of the multivariate data of *k* attributes. It utilizes pre-tuned *l* stacked layers of pre-tuned $n_{c}$ LSTM cells each. Then, it uses two dense NNs to smoothly transform the stacked LSTM output of $n_{c}$ values to $k\cdot n_{c}$ and finally to a $p_{l}\cdot k$ 1D array of consecutive values from the second NN layer. These values are then reshaped to form a 2D ($p_{l},k$) array. This array is the model’s final prediction output. That is, after passing through a batch normalization step to smooth outliers on the prediction value attributes.

The V-LSTM network, combined with BatchNormalization, was employed to help mitigate overfitting by stabilizing the learning process and introducing a mild regularization effect. Additionally, early stopping mechanisms were incorporated during training to halt the process when performance on the validation set ceased to improve, further preventing overfitting. In addition, random noise was introduced into the autoencoder data to enhance the model’s learning of more error-prone features. However, while these actions prevent overfitting, they do not substitute for explicit regularization techniques such as dropouts or L1 or L2 penalties.

Finally, the proposed V-LSTM model includes variable depth data training selection (variable *n* in Figure 7), variable length of forecasting outputs (variable *p* in Figure 7), and model parameter autotuning processes for the number of cells per layer and the number of layers. It is well-suited for tasks like fermentation process forecasting, sensor data prediction, or any sequential regression application involving fuzzy or biological systems.

## 4. Experimental Results and Discussion

The following subsections present the authors’ experimentation and proposed model evaluation. Scenario I presents the authors’ experimentation with their implemented SmartBarrel IoT E-nose and E-torque devices. Scenario II presents the authors’ experimentation of their SmartBarrel measured fermentation data using a fuzzy controlled process trained by white wine fermentations, and Scenario III presents the evaluation of their proposed V-LSTM model for fermentation predictions, trained using a fuzzy autoencoding process.

### 4.1. Scenario I: Evaluation of the SmartBarrel E-Nose Device

The alcoholic fermentation of wine progresses through four characteristic phases. The initial lag phase (6–24 h) involves yeast acclimatization and minimal activity. This is followed by a transition to the vigorous exponential phase (3–7 days), where rapid sugar conversion occurs, producing most of the alcohol and CO_2_. The stationary phase (5–10 days) follows as yeast growth slows and flavors develop. Finally, the decline phase (7–14 days) completes fermentation as nutrients are depleted [77]. The maximum total duration typically ranges from 14 to 21 days (300–500 h), depending on yeast strain, temperature (typically between 12 and 32 °C), and sugar content.

In this experimental scenario, the E-nose SmartBarrel end device is attached to the fermenting wine at the end of the vigorous phase, where also the pomace removal from the must (yeast) has been completed (typically 1–3 days for white wines, 5–15 days for rose wines and 3–30 days for red wines). The grape variety used in this experimentation is the Debina Zitsa, Epirus, Greece, white grape variety, mixed with a small quantity of Vlachiko Zitsa, Epirus, Greece, red variety, and therefore vinified as a rosé wine. The tank used was a 75 L stainless steel tank filled with 50 L of yeast upon the first decantation of wine, which is primarily aimed at separating it from grape pomace, performed on the fifth day after the grape crushing process. Then, the E-nose lid was placed on top of the fermentation tank and air-tight sealed using the air pump at 1 atm. This process is illustrated in Figure 8.

Figure 8(1) illustrates the SmartBarrel prototype of the E-nose, with the lid placed on top of the fermenting wine (Figure 8(2)). Figure 8(3) illustrates the one-way release valve that releases the CO_2_ from the sealed tank. The lid is pressed against the tank using an air-pressurized rubber (see Figure 8 (5)). Finally, Figure 8(4) illustrates the E-nose controller device that connects to the winery Wi-Fi network and, from there, to the ThingBoard AS. Measurements of CO, CO_2_, and C2H5OH in tank gas concentrations, internal yeast temperature, and external room temperature are periodically sent to the cloud every $T_{p}$ minutes, which can be set at the ThingsBoard device dashboard. For this experiment, $T_{p}$ has been set to 5 min. Figure 9 shows these measurements for the wine four-phase fermentation interval. Figure 10 shows the temperature measurements of the wine (internal temperature) and the fermenting environment (temperature) during the fermentation phases. The mean internal temperature T=18 °C, while the environmental mean temperature is at 26 °C.

As shown in Figure 9d, gas analog values were acquired by the analog-to-digital decoder of 10-bit resolution of the E-nose control device and an Arduino UNO ATmega4809 8-bit AVR at 16 MHz equipped with 48 KB of Flash memory, 6 KB of SRAM, and 256 bytes of EEPROM. This device is also equipped with an onboard u-blox NINA-W102 Wi-Fi transponder and an IMU unit of the LSM6DS3TR module. Both Figure 9a–c follow similar trend line patterns, an indication of proper sensor calibration that did not lead to imbalances or erroneous fluctuations that are not illustrated in the corresponding real analog measurements. As shown in Figure 9c, the conversion from parts-per-million (ppm) to milligrams per liter (mg/L) for the ethanol (C2H5OH) concentration for the release in the air is transformed from gaseous ppm to liquid mg/L, according to Equation (Equation 23):(23)$$C_{\mathrm{ethanol}}=\left(\mathrm{ppm}\times \frac{M_{\mathrm{ethanol}}}{{\tilde{V}}_{m}}\right)\times \frac{1}{\alpha }$$
where C in mg/L is the alcohol mass concentration, ppm is the volume concentration (parts per million) of alcohol in the 5–7 cm air gap inside the tank, *M* is the molar mass of ethanol in (g/mol), which equals $M_{\mathrm{ethanol}}=46.07$, while $V_{m}=24.45$ is the molar volume of an ideal gas at standard conditions (L/mol at 25 °C and 1 atm). For nonstandard temperatures, it is calculated as $V_{m}=24.45\;L/\mathrm{mol}\times \frac{T}{298.15}$. Coefficient α≈0.00025 is the partition coefficient set for correction purposes. It expresses the distribution of ethanol between gas and liquid phases (see Equation (Equation 24)): (24)$$\alpha =\frac{C_{{C_{2}H_{5}OH}^{gas}}}{C_{{C_{2}H_{5}OH}^{liq}}}$$

During active wine fermentation phases, we expect gas traces of alcohol in the tank between 100 and 1000 ppm (0.5–5 mg/L) [78], so that the corresponding partitioning liquid ethanol part inside the fermenting wine should be between 10 and 150 g/L. The corresponding conversions from g/L and %Vol concentration are calculated based on Equation (Equation 25), and corresponding values are shown in Table 5.(25)$$\%\;\mathrm{vol}=\frac{C_{C_{2}H_{5}OH}}{\rho_{C_{2}H_{5}OH}}\times 100$$
where $C_{C_{2}H_{5}OH}$ is the ethanol concentration in (g/L) and $\rho_{C_{2}H_{5}OH}=0.789$ (g/mL) is the ethanol density at 20 °C.

As shown in Figure 9c, the alcohol curve follows the expected behavior, quadratically declining to zero alcohol gas inside the tank. Furthermore, the corresponding detected values of air alcohol vapors inside the fermentation tank match findings in the literature as expressed by Equation (Equation 18) and mentioned in the previous paragraph.

As shown in Figure 9b, there is a high frequency of CO_2_ bursts close to 10,000 ppm/burst for the first fermentation phases that start to reduce at the end of fermentation in terms of burst frequency and intensity for the mid-fermentation interval. However, a slight mean increase in CO_2_ gas concentrations remains in the tank at the decline fermentation phase between 1800 and 2500 ppm. This phenomenon is also indicated by the CO sensor that abruptly enters zero ppm during that phase (see Figure 9a). From Figure 9a, it is obvious that there are slight traces of CO during active fermentation phases with a mean of 12.5 ppm, which disappear in the decline fermentation phase.

### 4.2. Scenario II: Evaluation of the Fuzzy Controler

In this experimental scenario, a set of 25 fermentation curves have been used, provided by SmartBarrel debina fermentations and data generated to fit white wine fermentations as presented in [46], showing the fermentation of a white grape variety. The measurement included a time series of hourly tank fermentation data of temperature, pH, sugar concentration in g/L, measured alcohol concentrations in g/L, biomass concentrations in g/L, and CO_2_ in g/L. Since these measurements were measured hourly, a padding function was used to provide 5 min interval samples using interpolation. This dataset has been used to train the fuzzy alcohol concentration controller, as described analytically in Section 3.6.

This controller has been, in turn, used to infer alcohol concentrations of our rosé-white wine fermenting mixture of the Debina white variety and the Vlachiko red variety. The equipment used was the authors’ SmartBarell E-nose and E-tongue implementations. Figure 11 presents the actual and predicted values.

The experimental results present the actual measurements performed at the Debina–Vlachiko fermenting tank (of 50 L of fermenting wine), and the predicted line illustrates the predictions of the fuzzy controller. The green line is the finally calibrated fuzzy controller output with one additional step of a Gaussian filter added for stability. Upon calibration, the coefficient of determination achieved by the fuzzy controller was $R^{2}=0.87$, meaning that the controller’s temporal data alcohol inferences slightly declined from reality, specifically at the end of the stationary fermentation and death fermentation phases.

The authors denote the limitations of the small training dataset, adding to the divergence of the controller inferences (underfitting). Nevertheless, as a fuzzy inference process, it will be a handy tool for future fermentation of the Debina variety when adequate variety data are collected by the SmartBarrel implementation for the controller to train. The fact that the controller managed to follow the actual achievement of a significant $R^{2}$ score is a strong indication that it can be a significant tool for wineries to develop autoencoders that, in turn, can offer more sophisticated fermentation prediction tools like the V-LSTM predictor presented in Section 3.7. The following Section 4.3 evaluates the proposed V-LSTM model.

### 4.3. Scenario III: Evaluation of the V-LSTM Prediction Model

The V-LSTM model evaluation process involved 1200 alcoholic fermentations using the fuzzy encoder described in Section Fuzzy Fermentation Autoencoder over 21 fermentation days during which pH measurements, CO_2_, sugar concentration, total amount of alcohol concentrations, temperature, and biomass concentrations (enzymatic transformation content) were used as fuzzy wine fermentation parameters generated for all 1200 fermentations at a sampling rate of 5 min/parameters batch, which equates to 6048×1200×6 recorded data, approximately 7,000,000 time-annotated attribute vectors.

The training was performed on a standard cloud GPU of 4864 CUDA cores and 8 GB of RAM. That is the average CPU–GPU purchase potential that medium-sized wineries can achieve for minimal computational capabilities in monthly cloud rental costs. In such a case, the available generated data cannot be trained and loaded simultaneously, as it requires at least 128 GB of RAM. For this reason, the dataset has been partitioned into 12 chunks of 100 fermentations each. The number of chunks has been selectively chosen not to surpass the 7GB of memory required by the training process to load and train on each chunk. The testing dataset for the evaluation process has been selected as 20% of each chunked data. The training validation data split was also set to 20% per data chunk.

The training was carried out using a batch size of 32 for 100 epochs, of variable learning rate adaptation starting from $10^{-4}$ up to $10^{-10}$ with rate adaptation on constant epoch validation loss values at a reduction factor of 25% of the previous learning rate. Additionally, to maintain fault tolerance to vanishing gradients or training efforts of constant loss results, an early stopping mechanism has been instantiated, monitoring the validation loss function and storing the best model results, that is, the minimum RMSE values. This scenario splits the 21×1200 data into training and test sets. It has 20×12×6043 (attribute vectors) data during testing, which cannot be applied on a GPU as it needs 16 GB of GPU RAM. That is why the evaluation is performed on a 24-core virtual machine of x86-64bit CPU and 32 GB of RAM, prolonging training time.

#### 4.3.1. V-LSTM Hyperparameter Auto-Tuning

The V-LSTM model hyperparameter auto-tuning step has been performed using the fuzzy encoded dataset, first using the number of cells as a tuning parameter on a minimum of one LSTM-layer architecture, and testing inter-cell distances of ten randomly selected cell cases (unique cells random process) set to 32. The minimum cell value is 32, and the maximum is 512. The portion of the dataset used to validate the models was the $\frac{1}{12}$ of the original dataset, calculated similarly to RMSEV, as mentioned in [79]. The configuration of the RMSE loss function and the final validation loss at 10 training epochs has been examined for the minimal loss value cells. Figure 12 shows the RMSE loss over the number of cells per layer of the LSTM layer models examined.

From the cell tuning results in Figure 12, the $n_{c}$ = 64 cells/layer achieved the minimum RMSE loss of 10.53 for 10 training epochs, followed by the 480 cells/layer model of 10.53. Therefore, the number of 64 cells per layer has been automatically selected by the V-LSTM model. Next, the LSTM layer’s depth is tuned. Figure 13 shows the RMSE loss over the number of layers of the V-LSTM model examined. The tested LSTM models included a variable layer depth starting from 1–10 layers with a layer step of 10 and a random unique selection of 10 models to test keeping the optimal RMSE value number of cells as it was previously calculated. The portion of the dataset used to validate the models was the $\frac{1}{12}$ of the original dataset. The configuration of the RMSE loss function and the final validation loss at 10 training epochs have been examined for the minimal loss value layers.

In Figure 13, the LSTM layer tuning process indicated the minimum loss $n_{l}$ = 10, corresponding to the maximum number of testing layers, followed closely by $n_{l}$ = 4, by a separate validation set RMSE loss difference of 0.01. The authors also noticed that for portions of the dataset and a few epochs, the maximum number of layers always superseded all others, even at a minimal amount. Therefore, for the sake of significantly reducing the number of trainable parameters, they considered a penalty threshold expressed as the min–max loss value multiplied by the min–max weight expression of the number of layers *l*:(26)$$WRMSE_{Loss}^{l}=\frac{{\mathrm{RMSE}}_{l}-\min\left(\mathrm{RMSE}\right)}{\max(\mathrm{RMSE})-\min(\mathrm{RMSE})}\cdot \frac{l-\min(l)}{\max(l)-\min(l)}$$
where RMSE is calculated over a seperate validation set of fermentation parameters (RSMEV [79]).

However, in this case, the layer that achieved the minimum ${\mathrm{RMSE}}_{l}$ value was selected instead of the layer with the minimal $WRMSE_{Loss}^{l}$ value, and *l* = 10 layers were selected for the number of layers of the V-LSTM model. Table 6 shows the training parameters of the V-LSTM model. Two additional mechanisms are used: (1) an early stop mechanism that ends training when there is no improvement in validation losses for more than five epochs (early stopping patience) with a min_delta value of $10^{-5}$, and (2) an adaptive learning rate factor that reduces the default learning rate value of $10^{-4}$ using a 25% reduction factor down to $10^{-10}$, with a patience parameter set to 1 epoch.

During training of the V-LSTM model, the selected input temporal depth $n_{t}=288\times 5\;\min$ temporal measurements of the *k* = 6 sensory attribute data together have been transformed into a 2D array of 288×6 consecutive fermentation parameters. This array corresponds to the previous 24 h 5 min vectors of measurements of $X_{Bio}$, pH, T, $S_{sugar}$, $S_{CO_{2}}$ and $S_{alcohol}$ attributes. Upon data loading, the entire dataset is transformed into batches of (288, 6) two dimensional arrays. These inputs are processed through the first V-LSTM LSTM layer of pre-selected 64 cells out of four layers. The number 64 was chosen during hyperparameter tuning because it maintained the lowest RMSE value (see Figure 12).

#### 4.3.2. V-LSTM Evaluation Results

V-LSTM model training validation and evaluation results are presented in Figure 14, using the RMSE, while Table 7 summarizes the results. Initially, the experiment starts training on dataset-1 using the system’s GPU, maintaining a 7.2 GB of mean memory utilization during host-to-device transfers. Since an early stop mechanism is used, the number of training epochs per dataset varies. Figure 14a shows the achieved validation loss at the epoch end of each training dataset. Table 7 shows the number of epochs per dataset prior to an early stop if validation loss does not change for more than five epochs, as well as the total mean validation loss achieved by the V-LSTM prediction model. The validation RMSE is relatively low, suggesting the model learns well from the data. The minimal standard deviation indicates consistent performance across training folds, a strong sign of model stability.

Similarly, Figure 14b shows the mean evaluation RMSE loss achieved by the model. In total, 20% of the total dataset has been used for that purpose and evaluated using 24 core x86 CPU and 32 GB of system RAM. Evaluation loss varies between 0.1578 and 0.1610 over the trainable datasets, and its value at the training end of the last dataset (d12) was 0.1605. The mean RMSE achieved by the model was 0.1599. The evaluation RMSE is slightly lower than the validation RMSE. This indicates that the model does not overfit and generalizes well to unseen data. The even smaller standard deviation further confirms the model’s fair stability. Since the average number of epochs is around 41 (see Table 7), it is also an indication that the model was not overtrained and converged relatively quickly (41/100 epochs).

Four studies have been used to evaluate our V-LSTM predictor RMSE outputs. The first involves a classifier implemented using input sugar concentrations and alcohol degrees (alcohol concentrations) to predict fermentation behaviors over the first 72 h of the fermentation process [44]. This proposed model used two hidden NN layers that are only as big as the bottom part of our proposed V-LSTM predictor and achieved an accuracy of 80% in detecting normal and problematic fermentations using only one parameter as a predictor variable (alcoholic degrees), which also denotes the importance having dense number of samples to achieve significant accuracy results. From the above paper, the authors can only conclude that from the accuracies of this scale, the MSE loss parameters are significantly high and usually above 1.

The second study includes SVR modeling with a Gaussian kernel to predict pH values and sugar content (measured in Brix) of fermenting wines, which showed the minimum RMSE values achieved of 0.142 for pH and 0.804 for sugar content, giving a minimum RMSE mean value of 0.473 [45]. This result is 66.17% more than the loss results achieved by the V-LSTM model (V-LSTM loss is 66.17% less than 0.473).

The third study includes an NN model [46] of one hidden layer of up to 12 neurons to predict one of the fermentation measurements (preferably alcohol) having the other as model input (pH, CO_2_, $S_{s}ugar$, $X_{b}io$), which achieved a minimum MSE value of 0.99 and corresponds to an RMSE of 0.995, which is 83.92% more than the achieved RMSE values of the V-LSTM model (V-LSTM loss is 83.92% less than 0.995).

The last study includes an NN model with one hidden layer of 10 neurons that uses either randomly generated weights or weights set initially by a genetic algorithm for the prediction of alcohol and substrate concentrations [47]. The provided RMSE results for the genetic algorithm-defined initial weights are between 0.3 and 0.45. That is, a mean value of 0.37. The minimum achieved RMSE loss value of 0.3 is 46.67% more than the achieved RMSE values of the V-LSTM model (V-LSTM loss is 46.67% less than 0.3). The actual paper mentions in a sentence a testing loss of 0.03–0.045. However, Figures 10–15 from paper [47], that present actual alcohol and substrate concentrations, predicted values differ significantly from the actual values for a testing loss of this range. The loss is likely bigger than the loss of 0.3–0.45.

The SmartBarrel AI logic for forecasting wine fermentation parameters does not introduce regulatory considerations. From a regulatory perspective, AI technologies may fall under the European Food Safety Authority (EFSA), particularly when they impact on product safety, traceability, and labeling. For example, under EU Regulation (EC) No 178/2002 [80], food business operators are responsible for ensuring that automated systems do not compromise food safety or misinform consumers. The issues that may arise here are solely using the SmartBarrel IoT sensors rather than AI models since they only predict measurements and do not qualify or classify processes. Moreover, the SmartBarrel IoT devices issue is also surpassed since the fermentation processes are performed in small tanks as pilots for big fermentation processes, and the fermentation material by the SmartBarrel IoT devices is not distributed to the public but instead wasted as pilots coinciding in the same industrial area with larger qualified fermentation tanks or barrels. Under the ethical context, the SmartBarrel micro-fermentation processes augment rather than replace expert decision-making, protect data using centralized cloud repositories using non publicly available Wi-Fi data transmission channels maintained by the Industry, and secure HTTP and MQTT data transmissions. Therefore, a sustainable and socially responsible innovation is maintained.

## 5. Conclusions

This paper introduces a new wine fermentation monitoring system called SmartBarrel. The system is equipped with two low-cost IoT devices implementing an E-nose that monitors the release of fermenting gases such as alcohol, carbon monoxide (CO), and carbon dioxide (CO_2_), and an E-tongue that measures essential wine fermentation parameters, including fermentation residues, pH levels, sugar concentrations, and color changes. These IoT devices are installed on the lids of stainless steel fermentation tanks, where they continuously collect and transmit near-real-time measurements to the cloud. The system leverages the open-source community edition of the ThingsBoard platform and its mobile application for data visualization. Additionally, the proposed system supports fermentation forecasting and the ability to incorporate deep learning algorithms as cloud services, utilizing inputs from the Cassandra NoSQL database of sensory measurements and outputting classification incidents and predicted values via corresponding dashboards.

The authors also included the ability in their SmartBarrel system to offer an estimation of alcohol degrees without using an additional meter but with a fuzzy controller capable of inferring alcoholic content from fermentation parameters. This fuzzy controller can also be implemented at the IoT device level. The authors also proposed a cloud-based variable cells and layers LSTM model called V-LSTM for predicting future fermentation parameters trained on either past fermentation data or data provided by fuzzy logic autoencoders.

The SmartBarrel system data autoencoding process is very important since existing datasets for wine fermentations are very small in terms of measurement frequency, parameters, and sizes. However, appropriate control of the autoencoded data is needed to fit the actual experimental parameter curves. This autoencoding step is a preliminary mechanistic step used temporarily until sufficient fermentation data is acquired. The proposed V-LSTM model can also auto-tune its model schema of the number of LSTM layers and cells per layer used based on portions of data training and, therefore, selecting the best hyperparameters per case study or scenario. The authors set as future work a modification of their currently proposed V-LSTM model that can have multiple strands due to the selection of different hyperparameter values on re-training on new data, and therefore, policies to apply the best strand inferences for each case accordingly.

Experimentation focused on the E-nose and E-tongue SmartBarrel devices, validating their functionality as well as the overall SmartBarrel system capabilities of data logging and visualization. Furthermore, they experimented with their alcohol fuzzy controller inferences, achieving an $R^{2}$ score of 0.87 over new SmartBarrel fermentation data. The authors also experimented with their proposed V-LSTM predictor, showing that it can achieve RMSE scores down to 0.16, which is 46-84% less than existing prediction SVR and shallow NN models. The authors also denote the need for a dense-cloud-based measurement of wine fermenting parameters for deep learning models such as V-LSTM to achieve even better prediction results.

Given the dynamic and time-sensitive nature of fermentation processes, further evaluation of the forecasting outputs from the V-LSTM model and the fuzzy predictor is necessary. This evaluation should include a comparison of the fuzzy predictor with support vector regression (SVR)-trained predictors or even XGBoost-trained predictors, which can be programmed to operate at the device level and may outperform the fuzzy controller in terms of $R^{2}$ score and RMSE loss. Additionally, the introduction of temporal metrics, such as Dynamic Time Warping (DTW) and rolling window Mean Absolute Error (MAE), should be considered. Although their use as evaluation measures for deep learning models is limited in the literature, these metrics could still be employed to assess both the V-LSTM’s forecasting outputs and the fuzzy controller’s alcohol predictions. Such measures might enhance the models’ responsiveness to the various phases of fermentation and increase their confidence levels. The absence of this type of evaluation is a limitation of the current study.

The authors set as future work the validation extension of their SmartBarrel implementation and AI capabilities in small-scale fermentations of different Greek varieties performed and evaluated in the Enology lab of the Agricultural University of Athens, Greece in cooperation with the Grekis Inox company, Athens, Greece, improving their system’s technological readiness level and therefore introducing it as a low-cost product for automated small-scale vinification processes. Furthermore, testing of SmartBarrel IoT devices’ signal stability and interferences among their sensors and cables is part of future work, and extensive testing of the SmartBarrel E-nose and E-tongue prototypes as part of the finalization of an industry-ready, technological readiness level (TRL-9) product will be performed in cooperation with the Grekis company.

## Figures and Tables

**Figure 1 sensors-25-03877-f001:**
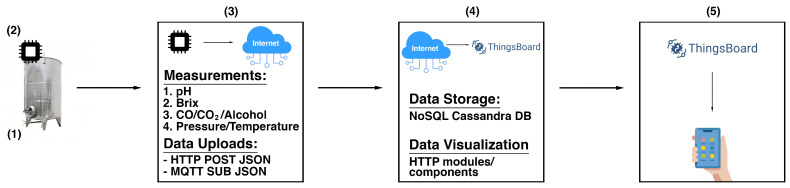
SmartBarrel high-level system architecture and system components and capabilities.

**Figure 2 sensors-25-03877-f002:**
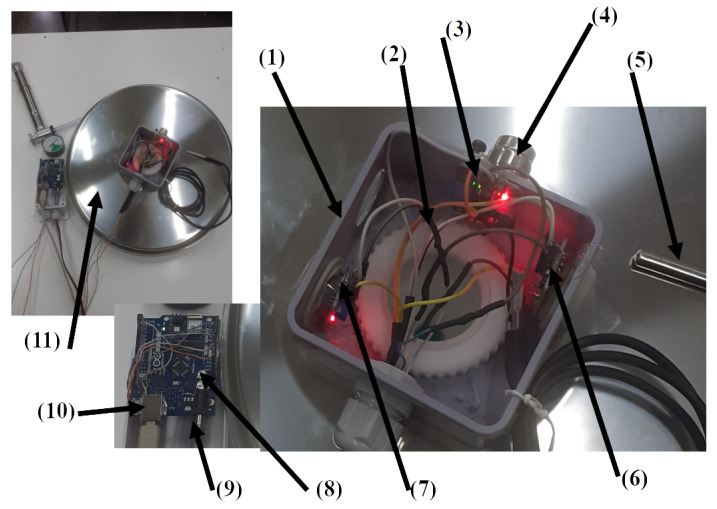
Illustration of the SmartBarrel E-nose device, parts, and sensory components.

**Figure 3 sensors-25-03877-f003:**
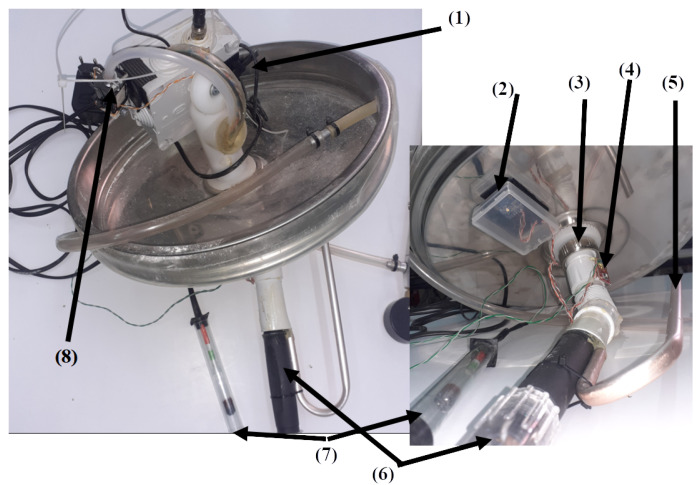
Illustration of the SmartBarrel E-tongue device, parts, and sensory components.

**Figure 4 sensors-25-03877-f004:**
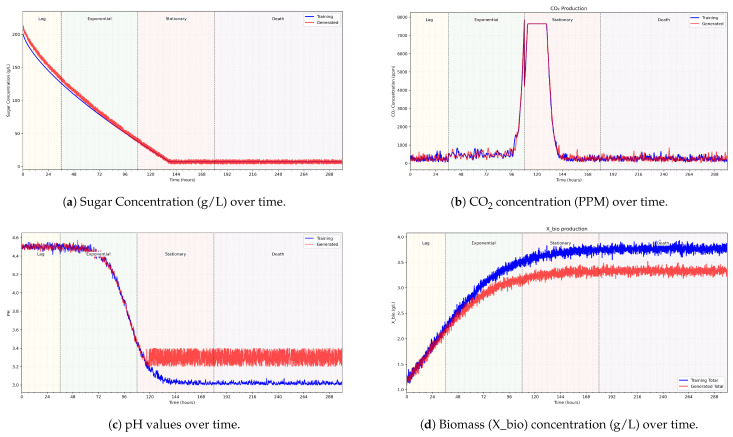
Fermentation parameters of a sample of generated data from the fuzzy autoencoder and actual data used for training provided by the SmartBarrel debina white wine fermentations and data generated following generated white wine fermentation curves as presented in [46]. The upper row displays (**a**) sugar concentration in g/L, and (**b**) carbon dioxide (CO_2_) concentrations in parts per million, while the lower row shows (**c**) pH values and (**d**) biomass concentration in g/L.

**Figure 5 sensors-25-03877-f005:**
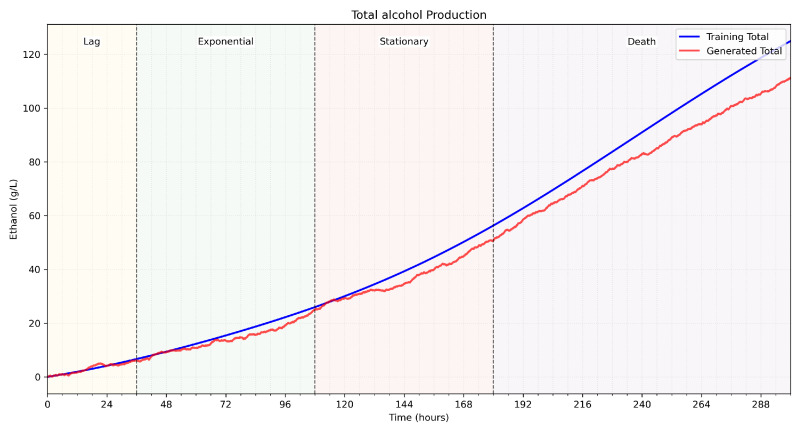
Alcohol concentration (g/L) over fermentation time of a sample of generated data from the fuzzy autoencoder, as calculated using the fuzzy inference alcohol prediction model.

**Figure 6 sensors-25-03877-f006:**
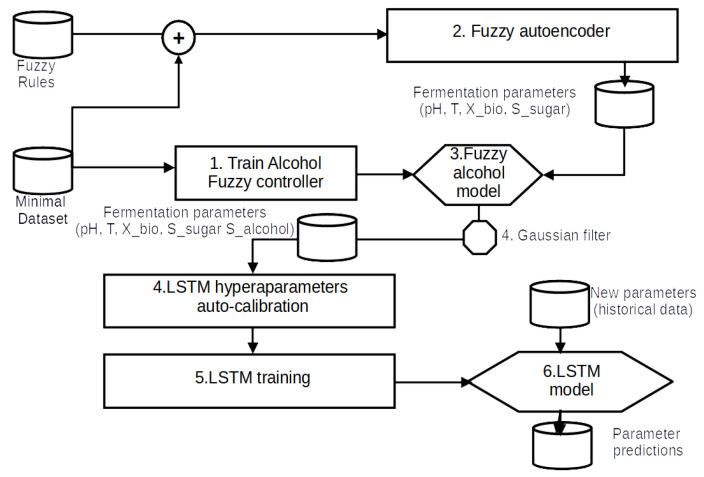
V-LSTM process flow diagram of fuzzy parameter generation, hyperparameter training, model training and predictions over new data streams.

**Figure 7 sensors-25-03877-f007:**
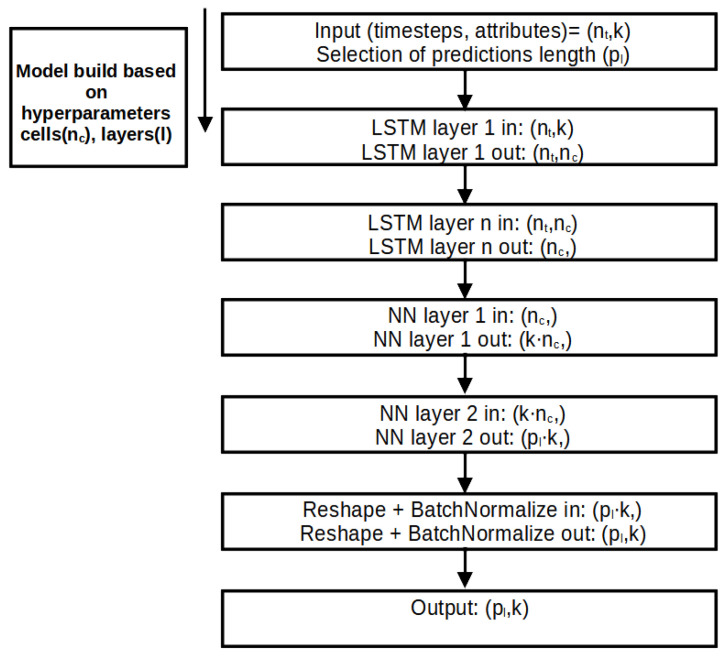
V-LSTM model structure, layers, data input and output.

**Figure 8 sensors-25-03877-f008:**
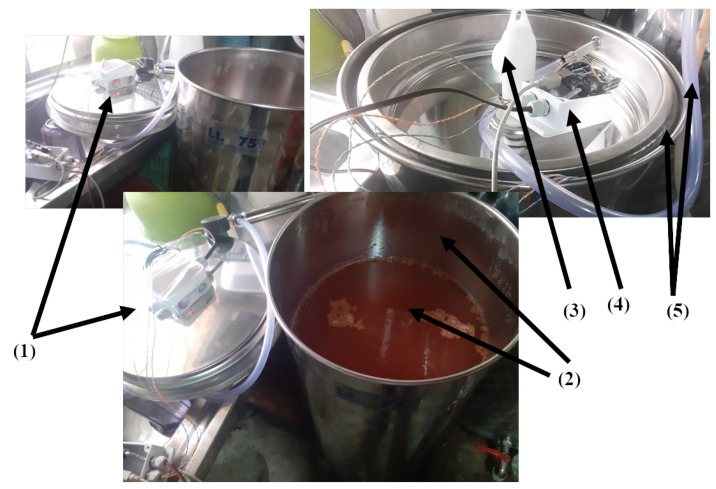
SmartBarrel E-nose evaluation for the fermentation of a Debina rosé mix of Debina white grape variety with a small quantity of Vlachiko red grape variety in the area of Zitsa, Epirus, Greece.

**Figure 9 sensors-25-03877-f009:**
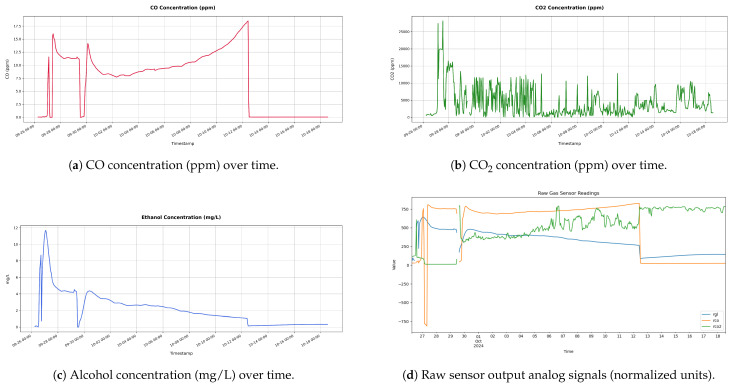
SmartBarrel E-nose gas concentration measurements from MQ-7 (CO) MOS sensor, MG-811 (CO_2_) NDIR like solid-state electrochemical sensor, and MQ-3 MOS alcohol sensor, showing both processed concentrations and raw signals. The upper row displays (**a**) carbon monoxide (CO) and (**b**) carbon dioxide (CO_2_) concentrations in parts per million, while the lower row shows (**c**) gas alcohol concentrations in mg/L and (**d**) the corresponding raw sensor outputs.

**Figure 10 sensors-25-03877-f010:**
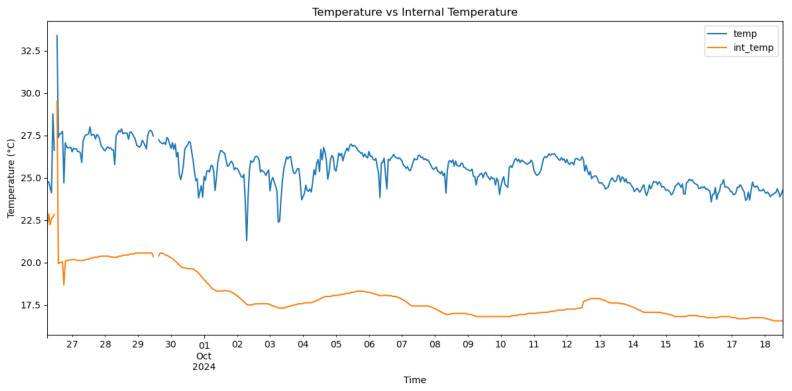
SmartBarrel E-nose temperature plot in the air and inside the fermenting yeast (internal temperature).

**Figure 11 sensors-25-03877-f011:**
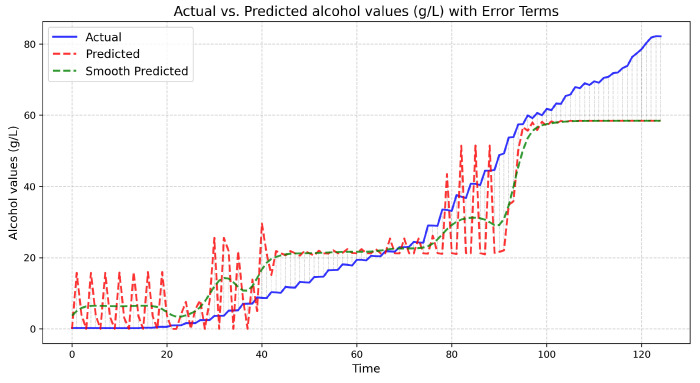
Alcohol fuzzy controller inference values (green line) over Debina–Vlachiko (Zitsa, Epirus, Greece) near-real-time fermentation curve of alcohol concentrations (blue line).

**Figure 12 sensors-25-03877-f012:**
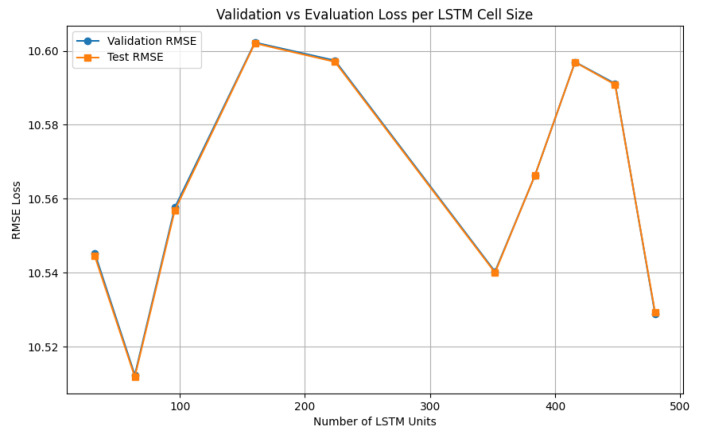
V-LSTM model tuning process of the number of cells per V-LSTM layer using the RMSE as loss function.

**Figure 13 sensors-25-03877-f013:**
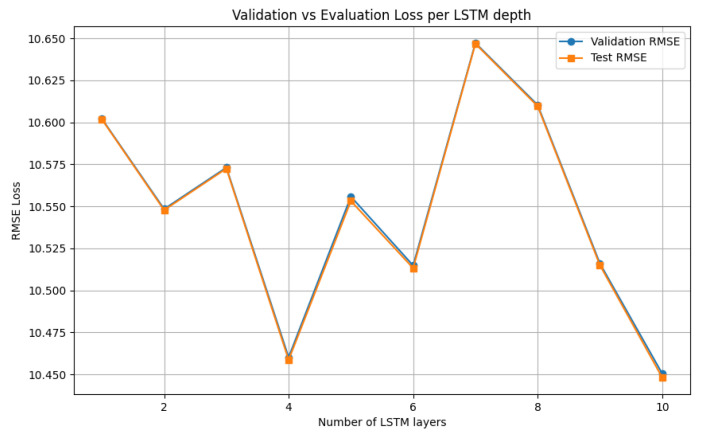
V-LSTM model tuning process of the number of model layers using the RMSE as loss function.

**Figure 14 sensors-25-03877-f014:**
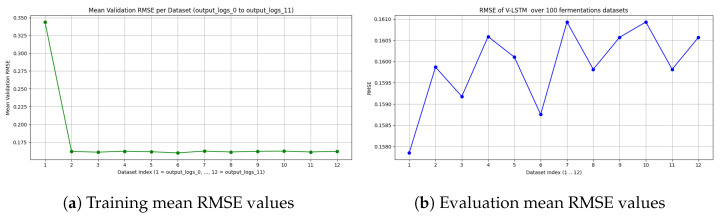
Validation and evaluation results of the V-LSTM model. (**a**) Training mean RMSE values over trainable fermentation datasets (d = 1…12) achieved at the last trainable epoch, and (**b**) Evaluation mean RMSE values over trainable fermentation datasets (d = 1…12) using 20% of the total trainable dataset (d = 1…12).

**Table 1 sensors-25-03877-t001:** Specifications of E-tongue and E-nose end node IoT devices and their sensors.

Device	Sensor	Connection	Min Value	Max Value	Resolution
E-tongue IoT device	pH meter *	Analog	0	14	0.1 pH
	Baume meter *	Analog	0 g/L	280 g/L	1.75 g/L
	Pressure sensor	I2C (TWI)	0 Bar	14 Bar	1 mBar
	Temperature sensor	OneWire	−40 °C	125 °C	0.1 °C
	RGB Sensor	I2C (TWI)	0	16 bit/channel	1 bit
E-nose IoT device	MQ-3 (Alcohol) *	Analog	0.4 mg/L	20 mg/L	∼0.018 mg/L
	MQ-7 (CO) *	Analog	10 ppm	2000 ppm	∼1 ppm
	MG811 (CO_2_) *	Analog	350 ppm	10,000 ppm	∼10 ppm
	Temperature sensor	OneWire	−40 °C	125 °C	0.1 °C

* MQ-3 and MQ-7 sensors were calibrated in clear air conditions, calculating base $R_{0}$ and voltage for MG811 at 400PPM (clear air) and then using the characteristic curves from the sensors’ datasheets provided. pH and Brix meters have been calibrated using different solutions of alcohol and sugar concentration solutions.

**Table 2 sensors-25-03877-t002:** Fuzzy controller uniform rules used for partitioning attributes (both fuzzy antecedents and consequents) into the three different classes (low, medium, high) with Gaussian MFs hyperparameters and calibrated values to achieve minimum RMSE.

Hyperparameter	Description	Calculation	Calibrated Value
$min(x_{i})$ offset $C_{min}$	offset from minimum value for the attribute $x_{i}$	$min\left(x_{i}\right)+C_{min}$	$-2\times min(x_{i})$
$max(x_{i})$ offset $C_{max}$	offset from minimum value for the attribute $x_{i}$	$max\left(x_{i}\right)+C_{max}$	$0.2\times max(x_{i})$
*w* curve bandwidth	Width of the gaussian-skewed gaussian curve expressed as a fraction of the $x_{i}$ standard deviation σ	$w=\frac{\sigma }{k}$	k=1…10 default k = 3
$MF_{c}^{low}$	center value of the low $x_{i}$ membership function	$min\left(x_{i}\right)-C_{\mathrm{off}}^{low}$	$C_{\mathrm{off}}^{low}=0$
$MF_{c}^{med}$	center value of the medium $x_{i}$ membership function	$\bar{x_{i}}+C_{\mathrm{off}}^{med}$	$C_{\mathrm{off}}^{med}=0$
$MF_{c}^{high}$	center value of the high $x_{i}$ membership function	$\mathrm{high}\left(x_{i}\right)+C_{\mathrm{off}}^{high}$	$C_{\mathrm{off}}^{high}=kd\cdot \delta$ kd = 1

**Table 3 sensors-25-03877-t003:** Fuzzy rule base for fermentation phases.

Fuzzy Antecedent	Fuzzy Consequent
IF phase is lag	THEN biomass grows slowly ($0.5(1-e^{-t/10})$)THEN sugar concentration remains high
IF phase is exponential	THEN biomass follows sigmoid ($3/(1+e^{-0.3(t-45)})$)THEN sugar concentration decays exponentially
IF phase is stationary	THEN biomass declines linearly (3.5−0.01(t−100))THEN sugar concentration approaches 30 g/L
IF phase is death	THEN biomass decays exponentially ($3e^{-0.005(t-336)}$)THEN sugar concentration stabilizes at 20 g/L
IF phase is lag	THEN pH = 4.5 (constant), T=25 ± 0.2°C
IF phase is exponential	THEN pH decreases sigmoidally, *T* strictly controlled
IF phase is stationary	THEN pH stabilizes near 3.2, *T* maintenance continues
IF phase is death	THEN pH slowly recovers, *T* control remains active

**Table 4 sensors-25-03877-t004:** Fuzzy rule-chain used by the fermentation autoencoding process.

Fuzzy Antecedents and Conditions	Fuzzy Consequent Conditions and Actions
If lag fermentation phase (0–20 h)	
Biomass:	$X_{bio}\left(t\right)=0.5\left(1-e^{-t/10}\right)+N\left(0,0.02^{2}\right)$
Sugar:	$S_{sugar}\left(t\right)=210+N\left(0,0.5^{2}\right)$ (constant high)
pH:	pH(t)=4.5±0.05 (no change)
Temp:	$T\left(t\right)=15+N\left(0,0.05^{2}\right)$ (strict control)
CO_2_:	$S_{CO_{2}}\left(t\right)=0.1X_{bio}\left(t\right)+E\left(0.1\right)$ (very low)
Alcohol:	$P_{alcohol}\left(t\right)=0$ (none produced)
If exponential fermentation phase (20–70 h)	
Biomass:	$X_{bio}\left(t\right)=0.5+\frac{3.5}{1+e^{-0.3(t-45)}}+N\left(0,0.05^{2}\right)$
Sugar:	$S_{sugar}\left(t\right)=210e^{-0.015t}+N\left(0,1^{2}\right)$
pH:	$pH\left(t\right)=4.5-\frac{1.8}{1+e^{-0.15(t-50)}}+N\left(0,0.05^{2}\right)$
Temp:	$T\left(t\right)=15+N\left(0,0.05^{2}\right)$
CO_2_:	$S_{CO_{2}}\left(t\right)=12X_{bio}\left(t\right)e^{-0.008{\left(t-60\right)}^{2}}+E\left(0.3\right)$
Alcohol:	$P_{alcohol}\left(t\right)=0.15X_{bio}\left(t\right)\left(1-0.3\sin\left(t/50\right)\right)$
If stationary fermentation phase (70–150 h)	
Biomass:	$X_{bio}\left(t\right)=3.5-0.01\left(t-100\right)+N\left(0,0.03^{2}\right)$
Sugar:	$S_{sugar}\left(t\right)=30+50e^{-0.02(t-100)}+N\left(0,0.8^{2}\right)$
pH:	pH(t)=3.2±0.1 (stabilized low)
Temp:	$T\left(t\right)=15+N\left(0,0.05^{2}\right)$
CO_2_:	$S_{CO_{2}}\left(t\right)=4X_{bio}\left(t\right)e^{-0.01{\left(t-120\right)}^{2}}+E\left(0.2\right)$
Alcohol:	$P_{alcohol}\left(t\right)=0.9X_{bio}\left(t\right)\left(1-0.1\sin\left(t/30\right)\right)$
If death fermentation phase (>150 h)	
Biomass:	$X_{bio}\left(t\right)=3.0e^{-0.005(t-200)}+N\left(0,0.05^{2}\right)$
Sugar:	$S_{sugar}\left(t\right)=20+N\left(0,0.3^{2}\right)$ (constant low)
pH:	$pH\left(t\right)=3.0+0.5e^{-0.01(t-150)}+N\left(0,0.05^{2}\right)$
Temp:	$T\left(t\right)=15+N\left(0,0.05^{2}\right)$
CO_2_:	$S_{CO_{2}}\left(t\right)=0.5X_{bio}\left(t\right)+E\left(0.1\right)$
Alcohol:	$P_{alcohol}\left(t\right)=0.8X_{bio}\left(t\right)$ (slow decline)

**Table 5 sensors-25-03877-t005:** Wine ethanol concentration conversion table.

Ethanol (g/L)	% vol
10	1.27
12.6	1.59 (Table wine minimum)
45	5.7
82.9	10.5 (Dry wine minimum)
94.7	12.0 (Typical non-dry wine)
150	19.0 (Fortified wine maximum)

**Table 6 sensors-25-03877-t006:** Summary of training parameters of the V-LSTM model.

Parameter	Value
Number of attributes (*k*)	6
Time window length ($n_{t}$)	255 (12 × 24)—24 h
Prediction length ($p_{l}$)	288 (12 × 24)—24 h
Number of LSTM layers (*l*)	10
Number of LSTM cells per layer ($n_{c}$)	64
Optimizer	Adam
Minimum learning rate	$1\times 10^{-10}$
Learning rate reduction factor	25%
Learning rate patience	1 epoch
Early stopping patience	5 epochs
Max number of epochs	100

**Table 7 sensors-25-03877-t007:** Mean and standard deviation of RMSE for V-LSTM evaluation (testing) and validation (train) over 12 datasets, each one representing 100 fermentations with 5 min resolution measurements.

Measure							Mean			Std. Dev.		
Validation RMSE (Epochs = 40.66 ≈41)							0.161468			0.001817		
Evaluation RMSE							0.159915			0.000895		
Trainable Epochs over train datasets												
Dataset	D1	D2	D3	D4t	D5	D6	D7	D8	D9	D10	D11	D12
Epochs	51	67	47	55	44	38	36	33	33	30	29	25

## Data Availability

The data presented in this study will be available or partly available on request from the corresponding author for research purposes only due to restrictions (privacy issues of a future market service or tool promoted and supported by the authors).

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
