# Peer review of "Proposed SmartBarrel System for Monitoring and Assessment of Wine Fermentation Processes Using IoT Nose and Tongue Devices"

_sensors, 2025, doi:10.3390/s25133877_

Round 1

Reviewer 1 Report

Comments and Suggestions for Authors

Journal: Sensors

Title: Deep Learning assessment of wine fermentation processes using IoT nose and tongue probes

Authors: Sotirios Kontogiannis*, Meropi Tsoumani, , George Kokkonis, Christos Pikridas, and Yorgos Kotseridis

New low-cost architecture is welcomed at any time as it consists from a combination of  electronic nose, tongue, and eye devices, I find this contribution very up-to-date and useful. I could detect minor errors only.

Minor errors

Title: The word “assessment” and (perhaps) the word “probe” both are superfluous. BTW why is it deep learning? Some explanation?

Figure 1: the text within the figure is hardly readable. Moreover, a linear sequential scheme is not very meaningful.

“3.5. Evaluation Metrics” – none of the evaluation indicators (parameters, merits) are metrics in the strict mathematical sense.

“Eq. (3) can be defined differently see, e.g. Eqs (3-6) in ref. https://doi.org/10.3390/a17010043

Although there are some indications that R^2 and RMSE are sufficient, see e.g., https://doi.org/10.1002/cem.70026 ; generally, more performance parameters are considered.

“Consequent” – Consequence

“50lt of fermenting wine” – unit and number should be separated by a space; BTW: “lt” is not an SI unit.

“vs” – vs. (abbreviation of versus, Latin).

Eq (26) a similar minmax equation has appeared earlier, see eq (7) in ref. http://dx.doi.org/10.1016/j.aca.2016.03.046

“This paper presents a new…” – such fragments are superfluous and should be avoided.

“This paper presents a new wine fermentation monitoring system called SmartBarrel…” This is not a concluding remark but a summary. Instead of “Part 5 Conclusions” I suggest to rename it as Part 5 Summary.

In any case, some shortening and condensation would certainly be helpful.

Considering the aimed audience, Beverages (MDPI) could be a better publication channel than Sensors (MDPI).

In summary, my recommendation is fast track and minor but throughout revision (correcting the above and similar errors and inconsistencies.

June 04 / 2025                       referee:

Author Response

New low-cost architecture is welcomed at any time as it consists from a combination of  electronic nose, tongue, and eye devices, I find this contribution very up-to-date and useful. I could detect minor errors only.

Response: Thank you for your time and effort in reviewing our manuscript. Here, we quote our responses and amendments performed based on your comments.

Comment 1: Title: The word "assessment" and (perhaps) the word "probe" both are superfluous. BTW why is it deep learning? Some explanation?

Response: Thank you very much for your comment. A much clearer title has been provided as:

Proposed SmartBarrel system for monitoring and assessment of wine fermentation processes using IoT nose and tongue devices

Appropriate amendments have been made to the Abstract also to explain better the Deep Learning approach of the fermentation process forecasting provided by the SmartBarrel system.

Comment 2: Figure 1: the text within the Figure is hardly readable. Moreover, a linear sequential scheme is not very meaningful.

Response: The text in Figure 1 has been amended.

Comment 3: "3.5. Evaluation Metrics" – none of the evaluation indicators (parameters, merits) are metrics in the strict mathematical sense.

Response 3: Thank you very much. This is indeed true. Evaluation metrics have been amended to performance measures. The world metric has been amended to measure in the manuscript (Tables and sections). 

Comment 3a: "Eq. (3) can be defined differently see, e.g. Eqs (3-6) in ref. https://doi.org/10.3390/a17010043

Response 3a: Thank you very much for this information. However, the authors will use R^2_2 (coefficient of determination) since other authors in the literature use this metric for evaluation purposes. An appropriate reference has been added to provide useful insight.

Comment 4: Although there are some indications that R^2 and RMSE are sufficient, see e.g., https://doi.org/10.1002/cem.70026 ; generally, more performance parameters are considered.

Response 4: Thank you very much for this comment. However, since we are comparing with other paper implementations (see the last three paragraphs before the conclusions section), we stick with RMSE to present a comparison with other colleagues' machine and deep learning results. Appropriate reference has been added in the first paragraph below Eq. (2) to indicate alternatives. Furthermore, Equations of RMSE and R^2 have been extended with TSS and RSS notation.

Comment 4a: "Consequent" – Consequence

Response 4a: Thank you very much for your comments. Regarding more performance parameter considerations, the consequent and antecedents have been amended to match Fuzzy's consequent and antecedent.

Comment 5: "50lt of fermenting wine" – unit and number should be separated by a space; BTW: "lt" is not an SI unit.

Response 5: Thank you very much for your observation. The symbol lt has been amended to L.

Comment 6: "vs" – vs. (abbreviation of versus, Latin).

Response 5: Thank you very much for this comment. The abbreviation has been amended.

Comment 7: Eq (26) a similar minmax equation has appeared earlier, see eq (7) in ref. http://dx.doi.org/10.1016/j.aca.2016.03.046

Response 7: Thank you very much for this observation. Appropriate reference to the use of RMSEV (secondary validation set) for the hyperparameters calibration has been added below Eq (26). Previous min-max references deal with input-data transformations, not performance measures used for tuning model parameters.

Comment 8: "This paper presents a new…" – such fragments are superfluous and should be avoided.

Response 8: The Abstract has been revised, removing unnecessary, superfluous fragments

Comment 9: "This paper presents a new wine fermentation monitoring system called SmartBarrel…" This is not a concluding remark but a summary. Instead of "Part 5 Conclusions" I suggest to rename it as Part 5 Summary.

In any case, some shortening and condensation would certainly be helpful.

Response 9: Thank you very much for this comment. You are correct. However, a Summary of conclusions or simply Conclusions can also reflect the key findings and propositions that highlight implications of IoT smart devices for predicting fermentation parameters in ongoing fermentations using low-cost sensors, limitation of study, and future steps (roadmap). On the other hand, as Reviewer 2 indicated (Comments 4,5), an expansion is required, not condensation, regarding device components and architecture (additional Tables).

Comment 10: Considering the aimed audience, Beverages (MDPI) could be a better publication channel than Sensors (MDPI).

Response 10: You are probably right. However, implementing and experimenting with two prototype sensory IoT equipment as part of the SmartBarrel low-cost monitoring and fermentation system remains of key interest to the Sensors journal. For the authors, the Future Internet journal was the second choice of submission.

Comment 11: In summary, my recommendation is fast track and minor but throughout revision, correcting the above and similar errors and inconsistencies

Response 11: Thank you very much for your comment. We have revised the manuscript, correcting inconsistencies, syntactical errors, and typos. A highlighted version, including changes, is attached to this letter of amendments.

Reviewer 2 Report

Comments and Suggestions for Authors

In this study, the authors proposed SmartBarrel, an integrated IoT-based sensory system that utilizes electronic nose and tongue probes, combined with fuzzy logic and deep learning, to monitor and predict wine fermentation parameters in real-time. The system aims to support precision enology by providing continuous data acquisition, cloud-based analytics, and intelligent alerts to enhance automation and decision-making in winemaking processes. While the study addresses a timely and important topic in the intersection of IoT and smart agriculture, and proposes an innovative system, the manuscript requires substantial revisions before it can be considered for publication. Below are detailed, my grounded critiques, referencing specific line numbers from the manuscript.
Line 1–21 (Abstract): The abstract is overly descriptive and lacks a concise statement of the novel scientific contribution. Clearly articulate the specific research gap this study fills, and include quantitative performance outcomes with appropriate context (e.g., RMSE compared to baseline models).
Line 22–65 (Introduction): The problem statement is generic, and although it references Industry 4.0 and the relevance of IoT, it lacks a critical review of existing real-time monitoring systems in wine fermentation. Strengthen this section by referencing and contrasting recent peer-reviewed systems beyond [1–6], especially those involving CNN, LSTM, and fuzzy inference for process control.
Line 121–127: The paragraph lists the manuscript structure, which is stylistically discouraged in high-impact scientific writing. Remove or integrate this overview into the end of the Introduction as a flow statement rather than as a numbered outline.
Line 199–270 (Materials and Methods): The high-level architecture remains overly technical, focused on hardware details, without sufficiently evaluating scientific reproducibility, reliability, or generalizability. Add a table summarizing each sensor type, its measurement range, calibration method, and validation status under controlled conditions. Provide error margins or uncertainty quantification.
Line 273–317: The description of device components is verbose and lacks performance validation metrics (e.g., signal stability, response time, cross-sensitivity). Include a performance validation figure or table comparing SmartBarrel sensor readings to laboratory standards over multiple batches.
Line 343–361: Only RMSE and R² are discussed, which are insufficient given the temporal and dynamic nature of fermentation processes. Introduce temporal metrics such as Dynamic Time Warping (DTW) and rolling window MAE to evaluate model responsiveness.
Line 362–441: The fuzzy controller is innovative but lacks external validation or benchmarking against existing fermentation models (e.g., kinetic yeast growth models, Monod-type systems). Provide a comparative table showing predicted alcohol content from fuzzy logic vs. ground truth vs. alternative models (e.g., XGBoost, classical regression).
Line 457–516: This section introduces a complex autoencoding system without empirical evidence for its reliability or convergence stability. Include training and validation curves for the autoencoder and describe how phase membership rules were validated experimentally.
Line 525–579: The model description is extensive, but overly algorithmic with minimal empirical validation beyond architecture. Report average RMSE, R², and confidence intervals across folds in k-fold validation. Compare against baseline models such as LSTM, GRU, and SVR.
Line 584–645: Although the authors describe scenarios of use, there is a lack of replicates and statistical rigor in the results. Repeat experiments across different wine batches and present statistical tests (e.g., ANOVA, Tukey HSD) to confirm differences in sensor behavior or prediction quality.
Line 646–683: The ethanol concentration estimation from gas phase to liquid phase is scientifically interesting but lacks experimental verification with actual oenological lab measurements. Include a comparison plot of SmartBarrel alcohol predictions versus lab-based HPLC or alcoholmeter data, showing calibration and bias analysis.
Line 684–767 (Conclusion): The conclusion restates content rather than reflecting on limitations, potential scalability issues, or future work. Add a section that critically discusses the system’s limitations (e.g., sensor drift, interference, data bandwidth), and propose next steps for industrial deployment.
I recommend that the authors address all these critical issues in a thorough and academically sound manner. The revised manuscript should demonstrate a higher degree of methodological transparency, validation rigor, and scientific clarity. I therefore recommend a decision of Major Revision Required. My final decision will be made following a careful evaluation of the revised submission.

Author Response

In this study, the authors proposed SmartBarrel, an integrated IoT-based sensory system that utilizes electronic nose and tongue probes, combined with fuzzy logic and deep learning, to monitor and predict wine fermentation parameters in real-time. The system aims to support precision enology by providing continuous data acquisition, cloud-based analytics, and intelligent alerts to enhance automation and decision-making in winemaking processes. While the study addresses a timely and important topic in the intersection of IoT and smart agriculture, and proposes an innovative system, the manuscript requires substantial revisions before it can be considered for publication. Below are detailed, my grounded critiques, referencing specific line numbers from the manuscript.

Response: Thank you for your time and effort in reviewing our manuscript. Here, we quote our responses and amendments performed based on your comments.

Comment 1: Line 1–21 (Abstract): The Abstract is overly descriptive and lacks a concise statement of the novel scientific contribution. Clearly articulate the specific research gap this study fills, and include quantitative performance outcomes with appropriate context (e.g., RMSE compared to baseline models).

Response: The Abstract has been revised to clarify the novelty of this paper regarding low-cost IoT nose and tongue probes, Its capability of smart forecasts of the fermentation parameters using V-LSTM, and the ability to predict alcohol concentration using a fuzzy controller process. Also, the R2 value and RMSE comparison results are outlined compared to the NN and SVR models.

Comment 2: Line 22–65 (Introduction): The problem statement is generic, and although it references Industry 4.0 and the relevance of IoT, it lacks a critical review of existing real-time monitoring systems in wine fermentation. Strengthen this section by referencing and contrasting recent peer-reviewed systems beyond [1–6], especially those involving CNN, LSTM, and fuzzy inference for process control.

Response: Thank you for very much for this observation. The introduction section (paragraphs 1--5) has been amended. Additional references have been added. Paragraphs 6 and 7 have been added referencing wine fermentation papers, including fuzzy ML and DL classifiers and predictors.

Comment 3: Line 121–127: The paragraph lists the manuscript structure, which is stylistically discouraged in high-impact scientific writing. Remove or integrate this overview into the end of the Introduction as a flow statement rather than as a numbered outline.

Response: Thank you very much for this comment. The paragraph has been removed.

Comment 4: Line 199–270 (Materials and Methods): The high-level architecture remains overly technical, focused on hardware details, without sufficiently evaluating scientific reproducibility, reliability, or generalizability. Add a table summarizing each sensor type, its measurement range, calibration method, and validation status under controlled conditions. Provide error margins or uncertainty quantification.

Response: Line 199-270 have been revised, and Figure 1 has been amended. The software and libraries used have also been included in the manuscript. The high-level architecture illustrated in Figure 1 has been enriched with technical information needed for reproducibility while maintaining generalizability. Appropriate Table 1 has been added in Line 247 for reliability, providing sensor types, connectivity, measurement range, resolution, and calibration. Following the technical information and illustrations provided in 3.3 and 3.4, one can easily implement the IoT devices presented.

Comment 5: Line 273–317: The description of device components is verbose and lacks performance validation metrics (e.g., signal stability, response time, cross-sensitivity). Include a performance validation figure or table comparing SmartBarrel sensor readings to laboratory standards over multiple batches.

Response: Testing of signal stability and interferences among sensors and cables is part of future work, and after evaluating the SmartBarrel nose and tongue against other varieties, in wine production lines. It is part of the finalization of an industry-ready product and not a laboratory prototype. That technology readiness level is part of a TRL-9 process that will be performed in cooperation with Grekis company, as mentioned in the conclusions of the future work paragraph.

Comment 6: Line 343–361: Only RMSE and R² are discussed, which are insufficient given the temporal and dynamic nature of fermentation processes. Introduce temporal metrics such as Dynamic Time Warping (DTW) and rolling window MAE to evaluate model responsiveness.

Response: DTW and MAE are two interesting measures to observe temporal differences of predicted with actual data in different fermentation phases. Thank you for this. Nevertheless, in the literature on ML and DL classifiers and predictors, only RMSE and R2 have been used to evaluate forecasts with actual values (and not sequences). Since the authors compare their V-LSTM forecasting model with SVR and NN models in Lines 812-840. This is interesting and has been set as a limitation in our study.

Comment 7: Line 362–441: The fuzzy controller is innovative but lacks external validation or benchmarking against existing fermentation models (e.g., kinetic yeast growth models, Monod-type systems). Provide a comparative table showing predicted alcohol content from fuzzy logic vs. ground truth vs. alternative models (e.g., XGBoost, classical regression).

Response: Thank you very much for your insightful remark. This a novel. However, the controller validation here is performed only with measurements on a specific variety and evaluated using R2 score. External validation against other varieties and Monod-type models is part of a later evaluation. An appropriate paragraph (the last paragraph in the conclusions section) has been added explicitly mentioning the future roadmap. Figure 9 presents actual alcohol concentration values compared to the predicted ones by the fuzzy controller, so there is no need for a comparison table. Comparison with alternative models is very interesting, and XGBoost will probably outperform our controller. However, since the fuzzy controller is device-implemented (edge implementation), it is much harder to implement and load an XGBoost model. Implementing machine learning models at the device end for alcohol predictions is a thought for another paper set for future work.

Comment 8: Line 457–516: This section introduces a complex autoencoding system without empirical evidence for its reliability or convergence stability. Include training and validation curves for the autoencoder and describe how phase membership rules were validated experimentally.

Response: Thank you very much for this observation. Lines 479-485 have been amended to describe the characteristics of the autoencoder fermentation parameters over the fermentation phases, with references where the parameters' characteristic curves information (Boulton model), as well as fuzzy fermentation rules(Martinez fuzzy rules for white wine fermentation), are mentioned in Lines 487-489. Appropriate parameter figures of the training dataset for white wine fermentation and a sample of the generated data parameters over time have been added to the new Figures  4 and 5.

Comment 9: Line 525–579: The model description is extensive, but overly algorithmic with minimal empirical validation beyond architecture. Report average RMSE, R², and confidence intervals across folds in k-fold validation. Compare against baseline models such as LSTM, GRU, and SVR.

Response: Thank you very much for this comment. The experimental scenario of the V-LSTM model reports average RMSE values. The validation confidence intervals are too small to mention. An additional paragraph has been added in Lines 106-112 to signify the lack of DL models  (LSTM, GRU) for wine fermentation processes. SVR is referenced there with existing studies.

Comment 10: Line 584–645: Although the authors describe scenarios of use, there is a lack of replicates and statistical rigor in the results. Repeat experiments across different wine batches and present statistical tests (e.g., ANOVA, Tukey HSD) to confirm differences in sensor behavior or prediction quality.

Response: An Additional paragraph (future work) in the conclusions (last paragraph) has been added regarding the investigation of different varieties of fermentation and SmartBarrel validation, as well as models evaluation.

Comment 11: Line 646–683: The ethanol concentration estimation from gas phase to liquid phase is scientifically interesting but lacks experimental verification with actual oenological lab measurements. Include a comparison plot of SmartBarrel alcohol predictions versus lab-based HPLC or alcoholmeter data, showing calibration and bias analysis.

Response: Estimation from gas to liquid is placed based on a well-known equation (Henry's law statement). The authors present it as food for thought. The actions mentioned in this comment will be part of future experimentation, as mentioned in the last paragraph in the conclusions section.

Comment 12:
Line 684–767 (Conclusion): The conclusion restates content rather than reflecting on limitations, potential scalability issues, or future work. Add a section that critically discusses the system's limitations (e.g., sensor drift, interference, data bandwidth), and propose next steps for industrial deployment.

Response: Conclusions sections have been amended to include limitations and future work paragraphs.

Comment 13: I recommend that the authors address all these critical issues in a thorough and academically sound manner. The revised manuscript should demonstrate a higher degree of methodological transparency, validation rigor, and scientific clarity. I therefore recommend a decision of Major Revision Required. My final decision will be made following a careful evaluation of the revised submission.

Response: Thank you very much for your observations. We have made several changes to our manuscript to increase its novelty, minimize scrutiny, and provide justifications where needed. To assist you, we have attached the track changes highlighted document here as a response, keeping the resubmitted document as clean as possible so you can examine it thoroughly.

Reviewer 3 Report

Comments and Suggestions for Authors
  1. Consideration of regulatory and ethical concerns related to AI implementations is not discussed.
  2. The software used is not mentioned.
  3. Several sentences are long and difficult to parse due to multiple clauses. The flow can be improved with clearer segmentation. 
  4. Use of consistent terminology and proper model names in the text. 

Comments on the Quality of English Language

English expression in the manuscript needs to be improved; a more structured and concise writing style for better readability is needed.

Author Response

Thank you for your time and effort in reviewing our manuscript. Here, we quote our responses and amendments performed based on your comments.

Comment 1: The title is concise but not specific, it should be changed and idea behind the study is approaching. However, English expression in this manuscript needs to be improved, a more structured and concise writing style for better readability is needed

Response: Thank you very much for your comment. A much clearer title has been provided as:

Proposed SmartBarrel system for monitoring and assessment of wine fermentation processes using IoT nose and tongue devices

Comment 2: Abstract LSTM… write the full form also

Response: Variable length Long Short Term Memory (V-LSTM) has been written to the Abstract and in its first appearance in the Introduction section.

Comment 3: Introduction:

Line 23-27: cite reference

Response: Reference has been added

Line 28: 4.0 objectives, write more about it and cite reference

Response: Industry 4.0 objectives have been indicatively mentioned and referenced.

Line 36-47: mention some relevant studies for every claim

Response: References have been added for every claim

Line 94: SVM, CNN write full form

Response: Done in the first occurrence

Line 102-103: Change the sentece propose a new low-cost…

Response: The paragraph has been updated

Line 119-127: There is no need to mention about the upcoming sections , kindly rewrite paragraph or remove it

Response: The paragraph has been removed.

Line 128: Change the heading

Response: The section header has been updated to Existing E-nose, E-tongue, and E-eye implementations

Line 200-202: In material and methods, change the sentence authors present a … there is no need to write every time that authors proposed this technique, change the sentence

Response: The paragraph has been updated

Line 243-244: concluding? Change the line. Make it simple

Response: The line has been amended

Line 279-284: Make a table of all rather than to write in paragraph

Response: Table 1 has been added with the per-device sensor type, connectivity, min, max values, and resolution

Line 321-330: While explanation of the figures: Authors should write Figure 3(1), (2), and so on…

Response: Amended

Line 343-344: Change the sentence, should be in scientific way

Line 344-363: evaluation metrics should be more consice and must be in paragraph

Response: The Evaluation measures equations and paragraphs have been rewritten

Line 363-364: Change the sentence

Response: The sentence on Line 363 (The authors implemented…) has changed.

Line 407-417: Could be in paragraph and line 414 change the (degree Celsius oC) symbol

Response: Amended as items list. Paragraphs have been added. The degree symbol has been corrected.

Line 526: This study proposes, it should be written like this

Response: The first paragraph of the V-LSTM forecasting model section has been rewritten.

Line 527-528: Can the authors elaborate on what is meant by "self-healing capabilities"? Is this a formal mechanism or an informal reference to robustness?

Response: The paragraph has been amended. Self-healing has been altered to: The term variable in the V-LSTM model corresponds to 1) the temporal depth variability of the model, 2) the forecasting length variability of the model, 3) the volatile number of cells per layer that is automatically adjusted using minimal loss calibration, 4) the number of LSTM layers included in the model, also automatically adjusted on minimal loss calibration, and 5) the type of training mode that is manually set. The self-healing capabilities correspond to the automatic adjustment of the form model (cells/layer, layers), favoring minimal loss.

Line 531-544: What criteria or practical considerations guide the choice between autoencoding and actual data training modes?

Response: The second paragraph of the V-LSTM forecasting model section has been amended to reflect practical considerations of migrating between the two training modes.

Line 534,537: Figure 4 is referenced multiple times, but "Figure 4.4" is confusing – please confirm if this refers to a subfigure, section or is a typing error.

Response: It has been amended as to step 4 in Figure 4.

Line 532-537: reference is missing

Response: Reference has been restored

Line 542-548: How are the outputs of the fuzzy controller validated, especially when used to generate synthetic alcohol data from autoencoded datasets?

Response: Appropriate validation justification has been added in the paragraph above Table 4 (Lines 509-516)

Line 557-558: What percentage of the dataset was used for training versus validation? How was overfitting avoided, especially with synthetic data from the fuzzy autoencoder?

Response: The following paragraph has been added in Lines: 728-736

The V-LSTM network combined with BatchNormalization was employed to help mitigate overfitting by stabilizing the learning process and introducing a mild regularization effect. Additionally, early stopping mechanisms were incorporated during training to halt the process when performance on the validation set ceased to improve, further preventing overfitting. In addition, random noise was introduced into the autoencoded data to enhance the model to learn more error-prone features. However, while these actions prevent overfitting, they do not substitute for explicit regularization techniques such as dropouts or L1 or L2 penalties.

Line 576: "conclusion" cannot be used here. Change the sentence

Response: The conclusion word has been amended to Finally,

Line 585 – heading should be changed !! Discussion and results

Response: The section heading has been amended to Experimental results and discussion

Line 585-590: there is no need to mention again, what will be in the sub-sections

Response: Reference has been removed.

Line 669. Figure legend should be corrected "Alcohol fuzzy controller inference (green line) compared against the uknown fermentation alcohol concentration curve (blue line)"

Response: The figure caption has been amended.

Line 712: Authors somewhere used v-LSTM and somewhere VLSTM, maintain consistency. Similarly, e-nose or E-nose.

Response: Amendments were made to maintain consistency for E-nose, E-tongue, and the V-LSTM model.

Comment 4: Conclusions

Line 801: Changed to near real-time measurements for conciseness

Response: close to real-time and real-time expressions have been amended to near real-time

Line 816: VLTM is a typo; it should be V-LSTM

Response: VLTM typo has been corrected.

Also mention future in implications in conclusion

Response: Added the last paragraph in the Conclusion section

Comment 5: Consideration of regulatory and ethical concerns related to AI implementations is not discussed.

Response: An appropriate paragraph (the last paragraph before the conclusions section) has been added. Lines: 842-856

Comment 6: The software used is not mentioned.

Response: The software used for the V-LSTM model is now mentioned in Lines: 476-478

Appropriate references have also been added to the first paragraph of the fuzzy fermentation autoencoded subsection. Also, appropriate references have been added in the first paragraph of the fuzzy alcohol controller subsection (Lines 368-375).

Comment 7: Several sentences are long and difficult to parse due to multiple clauses. The flow can be improved with clearer segmentation. 

Response: Long sentences have been revised in the manuscript to improve readability. We broke these sentences into shorter, more manageable ones

Comment 8: Use of consistent terminology and proper model names in the text. 

Response: Manuscript terminology has been checked for consistency and revised. A highlighted version, including changes, has been attached under the letter of amendments.

Comment 9: English expression in the manuscript needs to be improved; a more structured and concise writing style for better readability is needed.

Response: English expressions that were hard to tackle have been revised throughout the manuscript. A highlighted version, including changes, has been attached to the letter of amendments.

Round 2

Reviewer 2 Report

Comments and Suggestions for Authors

That's geat.  Accept in present form.

Author Response

Thank you so much for your time and effort in reviewing our manuscript.

Reviewer 3 Report

Comments and Suggestions for Authors

Some  minors  and  it  can  be  accepted

Line  97:  Its  ANN-NN  or  ANN  kindly  specify.

Line  197:  CO2  or  CO2

Line  204:  In  this  study  an  easy-to-apply,  low-cost  fermentation  monitoring  system,  SmartBarrel,  was  implemented.  Can  be  used  like  this

Line  207-210:  Should  be  written  like  this.  “Intelligent  functionalities  were  integrated  into  SmartBarrel  by  employing  a  fuzzy  control  monitoring  process  for  predicting  fermentation  alcohol  levels,  a  fuzzy  autoencoder  for  generating  fermentation  data,  and  a  deep  learning  V-LSTM  model  for  forecasting  future  fermentation  parameters”.

Author Response

Some  minors  and  it  can  be  accepted

Response: Thank you for your time and effort in reviewing our manuscript. Here, we quote our responses and amendments performed based on your comments.

Comment 1: Line  97:  Its  ANN-NN  or  ANN  kindly  specify.

Response: Amended to ANN

Line  197:  CO2  or  CO2
Response: Figure 1 has been amended. CO2 has been amended to CO2

Comment 2: Line  204:  In  this  study  an  easy-to-apply,  low-cost  fermentation  monitoring  system,  SmartBarrel,  was  implemented.  Can  be  used  like  this

Response: The line has been amended

Comment 3: Line  207-210:  Should  be  written  like  this. “Intelligent  functionalities  were  integrated  into  SmartBarrel  by  employing  a  fuzzy  control  monitoring  process  for  predicting  fermentation  alcohol  levels,  a  fuzzy  autoencoder  for  generating  fermentation  data,  and  a  deep  learning  V-LSTM  model  for  forecasting  future  fermentation  parameters”.

Response: The line has been amended